# Regret Minimization for Reinforcement Learning by Evaluating the Optimal Bias Function

**Zihan Zhang**
Tsinghua University
zihan-zh17@mails.tsinghua.edu.cn

**Xiangyang Ji**
Tsinghua University
xyji@tsinghua.edu.cn

## Abstract

We present an algorithm based on the *Optimism in the Face of Uncertainty* (OFU) principle which is able to learn Reinforcement Learning (RL) modeled by Markov decision process (MDP) with finite state-action space efficiently. By evaluating the state-pair difference of the optimal bias function $h^*$, the proposed algorithm achieves a regret bound of $\tilde{O}(\sqrt{SAHT})$[1] for MDP with $S$ states and $A$ actions, in the case that an upper bound $H$ on the span of $h^*$, i.e., $sp(h^*)$ is known. This result outperforms the best previous regret bounds $\tilde{O}(S\sqrt{AHT})$[Fruit et al., 2019] by a factor of $\sqrt{S}$. Furthermore, this regret bound matches the lower bound of $\Omega(\sqrt{SAHT})$[Jaksch et al., 2010] up to a logarithmic factor. As a consequence, we show that there is a near optimal regret bound of $\tilde{O}(\sqrt{SADT})$ for MDPs with a finite diameter $D$ compared to the lower bound of $\Omega(\sqrt{SADT})$[Jaksch et al., 2010].

## 1 Introduction

In this work we consider the Reinforcement Learning (RL) problem [Burnetas and Katehakis, 1997, Sutton and Barto, 2018] of an agent interacting with an environment. The problem is generally modelled as a discrete Markov Decision Process (MDP)[Puterman, 1994]. The RL agent needs to learn the underlying dynamics of the environment in order to make sequential decisions. At step $t$, the agent observes current state $s_t$ and chooses an action $a_t$ based on the policy learned from the past. Then the agent receives a reward $r_t$ from the environment, and the environment transits to state $s_{t+1}$ according to the states transition model. Particularly, both $r_t$ and $s_{t+1}$ are independent of previous trajectories, and are only conditioned on $s_t$ and $a_t$. In the online framework of reinforcement learning, we aim to maximize cumulative reward. Therefore, there is a trade-off between *exploration* and *exploitation*, i.e., taking actions we have not learned accurately enough and taking actions which seem to be optimal currently.

The solutions to exploration-exploitation dilemma can mainly be divided into two groups. In the first group, the approaches utilize the *Optimism in the Face of Uncertainty* (OFU) principle [Auer et al., 2002]. Under OFU principle, the agent maintains a confident set of MDPs and the underlying MDP is contained in this set with high probability. The agent executes the optimal policy of the best MDP in the confidence set [Bartlett and Tewari, 2009, Jaksch et al., 2010, Maillard et al., 2011, Fruit et al., 2018a]. In the second group, the approaches utilize posterior sampling [Thompson, 1933]. The agent maintains a posterior distribution over reward functions and transition models. It samples an MDP and executes corresponding optimal policy in each epoch. Because of simplicity and scalability, as well as provably optimal regret bound, posterior sampling has been getting popular in related research field [Osband et al., 2013, Osband and Van Roy, 2016, Agrawal and Jia, 2017, Abbasi-Yadkori, 2015].

## 1.1 Related Work

In the research field of regret minimization for reinforcement learning, Jaksch et al. [2010] showed a regret bound of $\tilde{O}(DS\sqrt{AT})$ for MDPs with a finite diameter $D$, and proved that it is impossible to reach a regret bound smaller than $\Omega(\sqrt{SADT})$. Agrawal and Jia [2017] established a better regret bound of $\tilde{O}(D\sqrt{SAT})$ by posterior sampling method. Bartlett and Tewari [2009] achieved a regret bound of $\tilde{O}(HS\sqrt{AT})$ where $H$ is an input as an upper bound of $sp(h^*)$. Fruit et al. [2018b] designed a practical algorithm for the constrained optimization problem in REGAL.C [Bartlett and Tewari, 2009], and obtained a regret bound of $\tilde{O}(H\sqrt{\Gamma SAT})$ where $\Gamma \leq S$ is the number of possible next states. On the other hand, Ouyang et al. [2017] and Theocharous et al. [2017] designed posterior sampling algorithms with Bayesian regret bound of $\tilde{O}(HS\sqrt{AT})$, with the assumption that elements of support of the prior distribution have a consistent upper bound $H$ for their optimal bias spans. Talebi and Maillard [2018] showed a problem-dependent regret bound of $\tilde{O}(\sqrt{\sum_{s,a} V(P_{s,a}, h^*)ST})$. Recently, Fruit et al. [2019] presented improved analysis of UCRL2B algorithm and obtained a regret bound of $\tilde{O}(S\sqrt{DAT})$.

There are also considerable work devoted to studying finite-horizon MDP. Osband and Van Roy [2016] presented PRSL to establish a Bayesian regret bound of $\tilde{O}(H\sqrt{SAT})$ using posterior sampling method. And later Azar et al. [2017] reached a better regret bound of $\tilde{O}(\sqrt{SAHT})$. Recently, Kakade et al. [2018] and Zanette and Brunskill [2019] achieved the same regret bound of $\tilde{O}(\sqrt{SAHT})$ by learning a precise value function to predict the best future reward of current state.

We notice a mistake about concentration of average of independent multinoulli trials in the proof of [Agrawal and Jia, 2017] (see Appendix.A for further details). This mistake suggests that they may not reduce a factor of $\sqrt{S}$ in their regret bounds.

## 1.2 Main Contribution

In this paper, we design an OFU based algorithm, and achieve a regret bound of $\tilde{O}(\sqrt{SAHT})$ given an upper bound $H$ on $sp(h^*)$. As a corollary, we establish a regret bound of $\tilde{O}(\sqrt{SADT})$ for the MDPs with finite diameter $D$. Meanwhile the corresponding lower bounds for the above two upper bounds are $\Omega(\sqrt{SAHT})$ and $\Omega(\sqrt{SADT})$ respectively. In a nutshell, our algorithm improves the regret bound by a factor of $\sqrt{S}$ compared to the best previous known results.

**Our Approach:** we consider regret minimization for RL by evaluating state-pair difference of the optimal bias function. Firstly, we observe that we can achieve a near-optimal regret bound with guide of the optimal bias function. Considering the fact that it is hard to estimate the optimal bias function directly [Ortner, 2008], we design a confidence set $\mathcal{H}_k$ of the optimal bias function. Based on $\mathcal{H}_k$ we obtain a tighter confidence set of MDPs and a better regret bound. It is notable that the order of samples in the trajectory is crucial when computing $\mathcal{H}_k$ in our algorithm, while it is ignored in previous methods. In this way, we utilize more information about the trajectory when computing the confidence set, which enables us to achieve a better regret bound.

## 2 Preliminaries

We consider the MDP learning problem where the MDP $M = \langle \mathcal{S}, \mathcal{A}, r, P, s_1 \rangle$. $\mathcal{S} = \{1, 2, ..., S\}$ is the state space, $\mathcal{A} = \{1, 2, ..., A\}$ is the action space, $P : \mathcal{S} \times \mathcal{A} \to \Delta^S$ [2] is the transition model, $r : \mathcal{S} \times \mathcal{A} \to \Delta^{[0,1]}$ is the reward function, and $s_1$ is the initial state. The agent executes action $a$ at state $s$ and receives a reward $r(s, a)$, and then the system transits to the next state $s'$ according to $\mathbb{P}(\cdot|s, a) = P_{s,a}$. In this paper, we assume that $\mathbb{E}[r(s, a)]$ is known for each $(s, a)$ pair, and denote $\mathbb{E}[r(s, a)]$ as $r_{s,a}$. It is not difficult to extend the proof to the original case.

In the following sections, we mainly focus on weak-communicating (see definition [Bartlett and Tewari, 2009]) MDPs.

**Assumption 1.** *The underlying MDP is weak-communicating .*

We first summarize several useful known results for MDPs and RL.

**Definition 1** (Policy). *A policy $\pi : \mathcal{S} \to \Delta^{\mathcal{A}}$ is a mapping from the state space to all distributions on the action space. In the case the support of $\pi(s)$ is a single action, we also denote this action as $\pi(s)$.*

Given a policy $\pi$, transition model $P$ and reward function $r$, we use $P_\pi$ to denote the transition probability matrix and $r_\pi$ to denote the reward vector under $\pi$. Specifically, when $\pi$ is a deterministic policy, $P_\pi = [P_{1,\pi(1)}, ..., P_{s,\pi(s)}]$ and $r_\pi = [r_{1,\pi(1)}, ..., r_{S,\pi(S)}]^T$.

**Definition 2** (Average reward). *Given a policy $\pi$, when starting from $s_1 = s$, the average reward is defined as:*

$$\rho_\pi(s) = \lim_{T \to \infty} \frac{1}{T} \mathbb{E}_{a_t \sim \pi(s_t), 1 \le t \le T} [\sum_{t=1}^{T} r_{s_t, a_t} | s_1 = s].$$

The optimal average reward and the optimal policy are defined as $\rho^*(s) = \max_\pi \rho_\pi(s)$ and $\Pi^*(s) = \arg\max_\pi \rho_\pi(s)$ respectively. It is well known that, under Assumption 1, $\rho^*(s)$ is state independent, so that we write it as $\rho^*$ in the rest of the paper for simplicity.

**Definition 3** (Diameter). *Diameter of an MDP $M$ is defined as:*

$$D(M) = \max_{s,s' \in \mathcal{S}, s \ne s'} \min_{\pi: \mathcal{S} \to \Delta_{\mathcal{A}}} T^\pi_{s \to s'},$$

*where $T^\pi_{s \to s'}$ denotes the expected number of steps to reach $s'$ from $s$ under policy $\pi$.*

Under Assumption 1, it is known the optimal bias function $h^*$ satisfies that

$$h^* + \rho^* \mathbf{1} = \max_{a \in \mathcal{A}} (r_{s,a} + P_{s,a}^T h^*) \tag{1}$$

where $\mathbf{1} = [1, 1, ..., 1]^T$. It is obvious that if $h$ satisfies (1), then so is $h^* + \lambda \mathbf{1}$ for any $\lambda \in \mathbb{R}$. Assuming $h$ is a solution to (1), we set[3] $\lambda = -\min_s h_s$ and $h^* = h + \lambda \mathbf{1}$, then the optimal bias function $h^*$ is uniquely defined. Besides, the span operator $sp : \mathbb{R}^S \to \mathbb{R}$ is defined as $sp(v) = \max_{s,s' \in [S]} |v_s - v_{s'}|$.

**The reinforcement learning problem.** In reinforcement learning, the agent starts at $s_1 = s_{start}$, and proceeds to make decisions in rounds $t = 1, 2, ..., T$. The $\mathcal{S}$, $\mathcal{A}$ and $\{r_{s,a}\}_{s \in \mathcal{S}, a \in \mathcal{A}}$ are known to the agent, while the transition model $P$ is unknown to agent. Therefore, the final performance is measured by the cumulative regret defined as

$$\mathcal{R}(T, s_{start}) := T\rho^* - \sum_{t=1}^{T} r_{s_t, a_t}.$$

The upper bound for $\mathcal{R}(T, s_{start})$ we provide is always consistent with that of $s_{start}$. In the following sections, we use $\mathcal{R}(T, s_{start})$ to denote $\mathcal{R}(T)$ for simplicity.

## 3 Algorithm Description

### 3.1 Framework of UCRL2

We first revisit the classical framework of UCRL2 [Jaksch et al., 2010] briefly. As described in Algorithm 1 (*EBF*), there are mainly three components in the UCRL2 framework: *doubling episodes*, *building the confidence set* and *solving the optimization problem*.

**Doubling episodes**: The algorithm proceeds through episodes $k = 1, 2, ....$ In the $k$-th episode, the agent makes decisions according to $\pi_k$. The episode ends whenever $\exists(s, a)$, such that the visit count of $(s, a)$ in the $k$-th episode is larger than or equal to the visit count of $(s, a)$ before the $k$-th episode. Let $K$ be the number of episodes. Therefore, we can get that $K \le SA(\log_2(\frac{T}{SA}) + 1) \le 3SA \log(T)$ when $SA \ge 2$ [Jaksch et al., 2010].

**Building the confidence set:** At the beginning of an episode, the algorithm computes a collection of plausible MDPs, i.e., the confidence set $\mathcal{M}_k$ based on previous trajectory. $\mathcal{M}_k$ should be designed properly such that the underlying MDP $M$ is contained by $\mathcal{M}_k$ with high probability, and the elements

in $\mathcal{M}_k$ are closed to $M$. In our algorithm, the confidence set is not a collection of MDPs. Instead, we design a 4-tuple $(\pi, P'(\pi), h'(\pi), \rho(\pi))$ to describe a plausible MDP and its optimal policy.

**Solving the optimization problem:** Given a confidence set $\mathcal{M}$, the algorithm selects an element from $\mathcal{M}$ according to some criteria. Generally, to keep the optimality of the chosen MDP, the algorithm needs to maximize the average reward with respect to certain constraints. Then the corresponding optimal policy will be executed in current episode.

### 3.2 Tighter Confidence Set by Evaluating the Optimal Bias Function

REGAL.C [Bartlett and Tewari, 2009] utilizes $H$ to compute $\mathcal{M}_k$, thus avoiding the issues brought by the diameter $D$. Similar to REGAL.C, we assume that $H$, an upper bound of $sp(h^*)$ is known. We design a novel method to compute the confidence set, which is able to utilize the knowledge of the history trajectory more efficiently. We first compute a well-designed confidence set $\mathcal{H}_k$ of the optimal bias function, and obtain a tighter confidence set $\mathcal{M}_k$ based on $\mathcal{H}_k$.

On the basis of above discussion, we summarize high-level intuitions as below:

**Exploration guided by the optimal bias function:** Once the true optimal bias function $h^*$ is given, we could get a better regret bound. In this case we regard the regret minimization problem as $S$ independent multi-armed bandit problems. UCB algorithm with Bernstein bound [Lattimore and Hutter, 2012] provides a near optimal regret bound. However, we can not get $h^*$ exactly. Instead, a tight confidence set of $h^*$ also helps to guide exploration.

**Confidence set of the optimal bias function:** We first study what could be learned about $h^*$ if we always choose optimal actions. For two different states $s, s'$, suppose we start from $s$ at $t_1$, and reach $s'$ the first time at $t_2$ ($t_2$ is a stopping time), then we have $\mathbb{E}[\sum_{t=t_1}^{t_2-1}(r_t - \rho^*)]$[4]$= \delta^*_{s,s'} := h^*_s - h^*_{s'}$ by the definition of optimal bias function. As a result, $\sum_{t=t_1}^{t_2-1}(r_t - \rho^*)$ could be regarded as an unbiased estimator for $\delta^*_{s,s'}$. Based on concentration inequalities for martingales, we have the following formal definitions and lemma.

**Definition 4.** *Given a trajectory* $\mathcal{L} = \{(s_t, a_t, s_{t+1}, r_t)\}_{1 \le t \le N}$, *for* $s, s' \in \mathcal{S}$ *and* $s \ne s'$, *let* $ts_1(\mathcal{L}) := \min\{\min\{t|s_t = s\}, N+2\}$. *We define* $\{ts_k(\mathcal{L})\}_{k \ge 2}$ *and* $\{te_k(\mathcal{L})\}_{k \ge 1}$ *recursively by following rules,*

$$te_k(\mathcal{L}) := \min\big\{\min\{t|s_t = s', t > ts_k(\mathcal{L})\}, N+2\big\},$$
$$ts_k(\mathcal{L}) := \min\big\{\min\{t|s_t = s, t > te_{k-1}(\mathcal{L})\}, N+2\big\}.$$

*The count of arrivals* $c(s, s', \mathcal{L})$ *from* $s$ *to* $s'$ *is defined as*

$$c(s, s', \mathcal{L}) := \max\{k|te_k(\mathcal{L}) \le N+1\}.$$

*Here we define* $\min \varnothing = +\infty$ *and* $\max \varnothing = 0$ *respectively.*

**Lemma 1** (Main Lemma)**.** *We say an MDP is flat if all its actions are optimal. Suppose* $M$ *is a flat MDP (without the constraint* $r_{s,a} \in [0, 1]$*). We run* $N$ *steps following an algorithm* $\mathcal{G}$ *under* $M$. *Let* $\mathcal{L} = \{(s_t, a_t, s_{t+1}, r_t)\}_{1 \le t \le N}$ *be the final trajectory. For any two states* $s, s' \in \mathcal{S}$ *and* $s \ne s'$, *let* $c(s, s', \mathcal{L})$, $\{te_k(\mathcal{L})\}_{k \ge 1}$ *and* $\{ts_k(\mathcal{L})\}_{k \ge 1}$ *be defined as in Definition 4. Then we have, for any algorithm* $\mathcal{G}$, *with probability at least* $1 - N\delta$, *for any* $1 \le c \le c(s, s', \mathcal{L})$ *it holds that*

$$\Big|\sum_{k=1}^{c}\Big(h^*_{s'} - h^*_s + \sum_{ts_k(\mathcal{L}) \le t \le te_k(\mathcal{L})-1}(r_t - \rho^*)\Big)\Big| \le (\sqrt{2N\gamma}+1)sp(h^*). \tag{2}$$

*where* $\gamma = \log(\frac{2}{\delta})$[5]

To use Lemma 1 to compute $\mathcal{H}_k$, we have to overcome two problems: (i) $M$ may not be *flat*; (ii) we do not have the value of $\rho^*$. Under the assumption the total regret is $\tilde{O}(HS\sqrt{AT})$, we can solve the problems subtly.

Let $reg_{s,a} = h^*_s + \rho^* - P^T_{s,a}h^* - r_{s,a}$, which is also called optimal gap [Burnetas and Katehakis, 1997] and could be regarded as the single step regret of $(s, a)$. Let $r'_{s,a} = h^*_s + \rho^* - P^T_{s,a}h^* = r_{s,a} + reg_{s,a}$

**Algorithm 1** EBF: Estimate the Bias Function
___
**Input:** $H, \delta, T$.
**Initialize:** $t \leftarrow 1, t_k \leftarrow 0$.
1: **for** episodes $k = 1, 2, ...$ **do**
2:     $t_k \leftarrow$ current time;
3:     $\mathcal{L}_{t_k-1} \leftarrow \{(s_i, a_i, s_{i+1}, r_i)\}_{1 \leq i \leq t_k-1}$;
4:     $\mathcal{M}_k \leftarrow BuildCS(H, \log(\frac{2}{\delta}), \mathcal{L}_{t_k-1})$;
5:     Choose $(\pi, P'(\pi), h'(\pi), \rho(\pi)) \in \mathcal{M}_k$ to maximize $\rho(\pi)$ over $\mathcal{M}_k$;
6:     $\pi_k \leftarrow \pi$;
7:     Follow $\pi_k$ until the visit count of some $(s, a)$ pair doubles.
8: **end for**
___

and $M' = \langle \mathcal{S}, \mathcal{A}, r', P, s_1 \rangle$. It is easy to prove that $M'$ is *flat* and has the same optimal bias function and optimal average reward as $M$. We attain by Lemma 1 that with high probability, it holds that

$$\left| \sum_{k=1}^{c(s,s',\mathcal{L})} \left( h_{s'}^* - h_s^* + \sum_{ts_k(\mathcal{L}) \leq t \leq te_k(\mathcal{L})-1} (r_{s_t,a_t} - \rho^*) \right) \right| \leq \sum_{t=1}^{N} reg_{s_t,a_t} + (\sqrt{2N\gamma}+1)sp(h^*). \quad (3)$$

Let $h' \in [0, H]^S$ be a vector such that (3) still holds with $h^*$ replaced by $h'$, then we can derive that

$$N_{s,a,s'} |(h_{s'}^* - h_s^*) - (h_{s'}' - h_s')| \leq 2 \sum_{t=1}^{N} reg_{s_t,a_t} + 2(\sqrt{2N\gamma}+1)H$$

where $N_{s,a,s'} := \sum_{t=1}^{N} \mathbb{I}[s_t = s, a_t = a, s_{t+1} = s'] \leq c(s, s', \mathcal{L})$. Because it is not hard to bound $\sum_{t=1}^{N} reg_{s_t,a_t} \approx \mathcal{R}(N)$ up to $\tilde{O}(HS\sqrt{AN})$ by REGAL.C [Bartlett and Tewari, 2009], we obtain that with high probability it holds

$$\hat{N}_{s,a,s'} |(h_{s'}^* - h_s^*) - (h_{s'}' - h_s')| = \tilde{O}(HS\sqrt{AN}). \quad (4)$$

As for the problem we have no knowledge about $\rho^*$, we can replace $\rho^*$ with the empirical average reward $\hat{\rho}$. Our claim about (4) still holds as long as $N(\rho^* - \hat{\rho}) = \tilde{O}(HS\sqrt{AN})$, which is equivalent to $\mathcal{R}(N) = \tilde{O}(HS\sqrt{AN})$.

Although it seems that (4) is not tight enough, it helps to bound the error term due to the difference between $h_k$ and $h^*$ up to $o(\sqrt{T})$ by setting $N = T$. (refer to Appendix.C.5.)

Based on the discussion above, we define $\mathcal{H}_k$ as:

$$\mathcal{H}_k := \{h \in [0, H]^S || L_1(h, s, s', \mathcal{L}_{t_k-1})| \leq 48S\sqrt{AT}sp(h) + (\sqrt{2\gamma T}+1)sp(h), \forall s, s', s \neq s'\}$$

where

$$L_1(h, s, s', \mathcal{L}) = \sum_{k=1}^{c(s,s',\mathcal{L})} \left( (h_{s'} - h_s) + \sum_{ts_k(\mathcal{L}) \leq i \leq te_k(\mathcal{L})-1} (r_i - \hat{\rho}) \right).$$

Together with constraints on the transition model (5)-(7) and constraint on optimality (8), we propose Algorithm 2 to build the confidence set, where

$$V(x, h) = \sum_s x_s h_s^2 - (x^T h)^2.$$

## 4 Main Results

In this section, we summarize the results obtained by using Algorithm 1 on weak-communicating MDPs. In the case there is an available upper bound $H$ for $sp(h^*)$, we have following theorem.

**Theorem 1** (Regret bound ($H$ known)). *With probability $1 - \delta$, for any weak-communicating MDP $M$ and any initial state $s_{start} \in \mathcal{S}$, when $T \geq p_1(S, A, H, \log(\frac{1}{\delta}))$ and $S, A, H \geq 20$ where $p_1$ is a polynomial function, the regret of EBF algorithm is bounded by*

$$\mathcal{R}(T) \leq 490\sqrt{SAHT \log(\frac{40S^2A^2T \log(T)}{\delta})},$$

---

**Algorithm 2** BuildCS($H, \gamma, \mathcal{L}$)

**Input:** $H, \gamma, \mathcal{L} = \{(s_i, a_i, s_{i+1}, r_i)\}_{1 \leq i \leq N}$

1: $\mathcal{H} \leftarrow \{h \in [0, H]^S | \ |L_1(h, s, s', \mathcal{L})| \leq 48S\sqrt{AT}sp(h) + (\sqrt{2\gamma T} + 1)sp(h), \forall s, s', s \neq s'\}$;
2: $N_{s,a} \leftarrow \max\{\sum_{t=1}^{N} \mathbb{I}[s_t = s, a_t = a], 1\}, \forall(s, a)$;
3: $\hat{P}_{s,a,s'} \leftarrow \frac{\sum_{t=1}^{N} \mathbb{I}[s_t=s, a_t=a, s_{t+1}=s']}{N_{s,a}}, \forall(s, a, s')$;
4: $\mathcal{O} \leftarrow \{\pi | \pi \text{ is a deterministic policy, and } \exists P'(\pi) \in \mathbb{R}^{S \times A \times S}, h'(\pi) \in \mathcal{H} \text{ and } \rho(\pi) \in \mathbb{R}, \text{such that}$

$$|P'_{s,a,s'}(\pi) - \hat{P}_{s,a,s'}| \leq 2\sqrt{\hat{P}_{s,a,s'}\gamma/N_{s,a}} + 3\gamma/N_{s,a} + 4\gamma^{\frac{3}{4}}/N_{s,a}^{\frac{3}{4}}, \tag{5}$$

$$|P'_{s,a}(\pi) - \hat{P}_{s,a}|_1 \leq \sqrt{14S\gamma/N_{s,a}} \tag{6}$$

$$|(P'_{s,a}(\pi) - \hat{P}_{s,a})^T h'(\pi)| \leq 2\sqrt{V(\hat{P}_{s,a}, h'(\pi))\gamma/N_{s,a}} + 12H\gamma/N_{s,a} + 10H\gamma^{3/4}/N_{k,s,a}^{3/4}, \tag{7}$$

$$P'_{s,\pi(s)}(\pi)^T h'(\pi) + r_{s,\pi(s)} = \max_{a \in \mathcal{A}} P'_{s,a}(\pi)^T h'(\pi) + r_{s,a} = h'(\pi) + \rho(\pi)\mathbf{1} \tag{8}$$

holds for any $s, a, s'\}$;
5: **Return:**$\{(\pi, P'(\pi), h'(\pi), \rho(\pi)) | \pi \in \mathcal{O}\}$.

---

*whenever an upper bound of the span of optimal bias function $H$ is known. By setting $\delta = \frac{1}{T}$, we get that $\mathbb{E}[\mathcal{R}(T)] = \tilde{O}(\sqrt{SAHT})$*

Theorem 1 generalizes the $\tilde{O}(\sqrt{SAHT})$ regret bound from the finite-horizon setting [Azar et al., 2017] to general weak-communicating MDPs, and improves the best previous known regret bound $\tilde{O}(H\sqrt{SAT})$[Fruit et al., 2019] by an $\sqrt{S}$ factor. More importantly, this upper bound matches the $\Omega(\sqrt{SAHT})$ lower bound up to a logarithmic factor.

Based on Theorem 1, in the case the diameter $D$ is finite but unknown, we can reach a regret bound of $\tilde{O}(\sqrt{SADT})$.

**Corollary 1.** *For weak-communicating MDP $M$ with a finite unknown diameter $D$ and any initial state $s_{start} \in \mathcal{S}$, with probability $1 - \delta$, when $T \geq p_2(S, A, D, \log(\frac{1}{\delta}))$ and $S, A, D \geq 20$ where $p_2$ is a polynomial function, the regret can be bounded by*

$$\mathcal{R}(T) \leq 491\sqrt{SADT(\log(\frac{S^3 A^2 T \log(T)}{\delta}))}.$$

*By setting $\delta = \frac{1}{T}$, we get that $\mathbb{E}[\mathcal{R}(T)] = \tilde{O}(\sqrt{SADT})$.*

We postpone the proof of Corollary 1 to Appendix.D.

Although *EBF* is proved to be near optimal, it is hard to implement the algorithm efficiently. The optimization problem in line 5 Algorithm 1 is well-posed because of the optimality equation (8). However, the constraint (7) is non-convex in $h'(\pi)$, which makes the optimization problem hard to solve. Recently, [Fruit et al. 2018b] proposed a practical algorithm SCAL, which solves the optimization problem in REGAL.C efficiently. We try to expand the *span truncation* operator $T_c$ to our framework, but fail to make substantial progress. We have to leave this to future work.

## 5 Analysis of EBF (Proof Sketch of Theorem 1)

Our proof mainly contains two parts. In the first part, we bound the probabilites of the bad events. In the second part, we manage to bound the regret when the good event occurs.

## 5.1 Probability of Bad Events

We first present the explicit definition of the bad events. Let $N_{s,a}^{(t)} = \sum_{i=1}^{t} \mathbb{I}[s_i = s, a_i = a]$. We denote $N_{k,s,a} = N_{s,a}^{(t_k-1)}$ as the visit count of $(s,a)$ before the $k$-th episode, and $v_{k,s,a}$ as the visit count of $(s,a)$ in the $k$-th episode respectively. We also denote $\hat{P}^{(k)}$ as the empirical transition model before the $k$-th episode.

**Definition 5** (Bad event). *For the $k$-th episode, define*

$$B_{1,k} := \left\{ \exists(s,a), s.t. |(P_{s,a} - \hat{P}_{s,a}^{(k)})^T h^*| > 2\sqrt{\frac{V(P_{s,a}, h^*)\gamma)}{\max\{N_{k,s,a}, 1\}}} + 2\frac{sp(h^*\gamma)}{\max\{N_{k,s,a}, 1\}} \right\},$$

$$B_{2,k} = \left\{ \exists(s,a,s'), s.t. |\hat{P}_{s,a,s'}^{(k)} - P_{s,a,s'}| > 2\sqrt{\frac{\hat{P}_{s,a,s'}^{(k)}\gamma}{\max\{N_{k,s,a}, 1\}}} + \frac{3\gamma}{\max\{N_{k,s,a}, 1\}} + \frac{4\gamma^{\frac{3}{4}}}{\max\{N_{k,s,a}, 1\}^{\frac{3}{4}}} \right\},$$

$$B_{3,k} = \left\{ |\sum_{1 \le t < t_k} (\rho^* - r_{s_t,a_t})| > 26HS\sqrt{AT\gamma}, \sum_{k' < k} \sum_{s,a} v_{k',s,a} reg_{s,a} > 22HS\sqrt{AT\gamma} \right\}$$

$$B_{4,k} = \left\{ \{(\pi^*, P^*, h^*, \rho^*) | \pi^* is\ a\ deterministic\ optimal\ policy\} \cap \mathcal{M}_k = \varnothing \right\}.$$

*The bad event in the $k$-th episode therefore is defined as $B_k = B_{1,k} \cup B_{2,k} \cup B_{3,k} \cup B_{4,k}$, and the total bad event $B$ is defined as $B := \cup_{1 \le k \le K+1} B_k$. At the same time, we have the definition of the good event as $G = B^C$.*

**Lemma 2** (Bound of $\mathbb{P}(B)$). *Suppose we run Algorithm 1 for $T$ steps, then $\mathbb{P}(B) \le (6AT + 12S^2A)SA \log(T)\delta$ when $T \ge A \log(T)$ and $SA \ge 4$.*

## 5.2 Regret when the Good Event Occurs

In this section we assume that the good event $G$ occurs. We use $\mathcal{R}_k$ to denote the regret in the $k$-th episode. We use $P_k'$, $P_k$, $\hat{P}_k$, $r_k$, $\rho_k$ and $h_k$ to denote $P_{\pi_k}'(\pi_k)$, $P_{\pi_k}$, $\hat{P}_{\pi_k}^{(k)}$, $r_{\pi_k}$, $\rho(\pi_k)$ and $h'(\pi_k)$ respectively. We define $v_k$ as the vector such that $v_{k,s} = v_{k,s,\pi_k(s)}, \forall s$, and introduce $\delta_{k,s,s'} = h_{k,s} - h_{k,s'}, \forall s, s'$.

Noting that for $\alpha > 0$, $\sum_k \sum_{s,a} v_{k,s,a} \frac{1}{\max\{N_{k,a,s}, 1\}^{\frac{1}{2}+\alpha}}$ could be roughly bounded by $O(T^{\frac{1}{2}-\alpha})$, which could be ignored when $T$ is sufficiently large. Therefore, we can omit such terms without changing the regret bound.

According to $B_{4,k}^C$ and the optimality of $\rho_k$ we have

$$\mathcal{R}_k = v_k^T(\rho^* \mathbf{1} - r_k) \le v_k^T(\rho_k \mathbf{1} - r_k) = v_k^T(P_k' - I)^T h_k$$
$$= \underbrace{v_k^T(P_k - I)^T h_k}_{①_k} + \underbrace{v_k^T(\hat{P}_k - P_k)^T h^*}_{②_k} + \underbrace{v_k^T(P_k' - \hat{P}_k)^T h_k}_{③_k} + \underbrace{v_k^T(\hat{P}_k - P_k)^T(h_k - h^*)}_{④_k}. \quad (9)$$

We bound the four terms in the right side of (9) separately.

**Term $①_k$** : The expectation of $①_k$ never exceeds $[-H, H]$. However, we can not directly utilize this to bound $①_k$. By observing that $①_k$ has a martingale difference structure, we have following lemma based on concentration inequality for martingales.

**Lemma 3.** *When $T \ge S^2 AH^2\gamma$, with probability $1 - 3\delta$, it holds that*

$$\sum_k ①_k \le KH + (4H + 2\sqrt{12TH})\gamma.$$

**Term $②_k$** : Recalling the definition of $V(x, h)$ in Section 3, $B_{1,k}^C$ implies that

$$②_k \le \sum_{s,a} v_{k,s,a} \left( 2\sqrt{\frac{V(P_{s,a}, h^*)\gamma}{\max\{N_{k,s,a}, 1\}}} + 2\frac{H\gamma}{\max\{N_{k,s,a}, 1\}} \right) \approx O\left( \sum_{s,a} v_{k,s,a}\sqrt{\frac{V(P_{s,a}, h^*)\gamma}{\max\{N_{k,s,a}, 1\}}} \right), \quad (10)$$

where $\approx$ means we omit the insignificant terms. We bound RHS of (10) by bounding $\sum_{s,a} N_{s,a}^{(T)} V(P_{s,a}, h^*)$ by $O(TH)$. Formally, we have following lemma.

**Lemma 4.** *When $T \geq S^2 A H^2 \gamma$, with probability $1 - \delta$*

$$\sum_{k,s,a} v_{k,s,a} \sqrt{\frac{V(P_{s,a}, h^*)\gamma}{\max\{N_{k,s,a}, 1\}}} \leq 21\sqrt{SAHT\gamma}.$$

**Term ③$_k$ :** According to (7) we have

$$③_k \leq \sum_{s,a} v_{k,s,a} L_2(\max\{N_{k,s,a}, 1\}, \hat{P}_{s,a}^{(k)}, h_k) \approx O\left(\sum_{s,a} v_{k,s,a} \sqrt{\frac{V(\hat{P}_{s,a}^{(k)}, h_k)\gamma}{\max\{N_{k,s,a}, 1\}}}\right) \qquad (11)$$

where $L_2(N, p, h) = 2\sqrt{V(p,h)\gamma/N} + 12H\gamma/N + 10H\gamma^{3/4}/N^{3/4}$. When dealing with the RHS of (11), because $h_k$ varies in different episodes, we have to bound the static part and the dynamic part separately. Noting that

$$\sqrt{V(\hat{P}_{s,a}^{(k)}, h_k)} - \sqrt{V(P_{s,a}, h^*)} \leq (\sqrt{V(\hat{P}_{s,a}^{(k)}, h_k)} - \sqrt{V(\hat{P}_{s,a}^{(k)}, h^*)}) + (\sqrt{V(\hat{P}_{s,a}^{(k)}, h^*)} - \sqrt{V(P_{s,a}, h^*)})$$

$$\leq \sqrt{|V(\hat{P}_{s,a}^{(k)}, h_k) - V(\hat{P}_{s,a}^{(k)}, h^*)|} + \sqrt{|V(\hat{P}_{s,a}^{(k)}, h^*) - V(P_{s,a}, h^*)|}$$

$$\leq \sqrt{4H \sum_{s'} \hat{P}_{s,a,s'}^{(k)} |\delta_{k,s,s'} - \delta_{s,s'}^*|} + \sqrt{4H^2|\hat{P}_{s,a}^{(k)} - P_{s,a}|_1}$$

$$\leq \sum_{s'} \sqrt{4H \hat{P}_{s,a,s'}^{(k)} |\delta_{k,s,s'} - \delta_{s,s'}^*|} + \sqrt{4H^2 \sqrt{\frac{14S\gamma}{\max\{N_{k,s,a}, 1\}}}}$$

$$\approx O\left(\sum_{s'} \sqrt{4H \hat{P}_{s,a,s'}^{(k)} |\delta_{k,s,s'} - \delta_{s,s'}^*|}\right),$$

$$(12)$$

According to the bound of the second term, it suffices to bound

$$\sqrt{H} \sum_{k,s,a} v_{k,s,a} \sum_{s'} \sqrt{\frac{\hat{P}_{s,a,s'}^{(k)} |\delta_{k,s,s'} - \delta_{s,s'}^*|}{\max\{N_{k,s,a}, 1\}}} \qquad (13)$$

Surprisingly, we find that this term is an upper bound for the fourth term.

**Term ④$_k$ :** Recalling that $\delta_{s,s'}^* = h_s^* - h_{s'}^*$, according to $B_{2,k}^C$ the fourth term can be bounded by:

$$④_k = \sum_{s,a} v_{k,s,a}(\hat{P}_{s,a}^{(k)} - P_{s,a})^T(h_k - h_{k,s}\mathbf{1} - h^* + h_s^*\mathbf{1}) = \sum_{s,a} v_{k,s,a} \sum_{s'} (\hat{P}_{s,a,s'}^{(k)} - P_{s,a,s})(\delta_{s,s'}^* - \delta_{k,s,s'})$$

$$\approx O\left(\sum_{s,a} v_{k,s,a} \sum_{s'} \sqrt{\frac{\hat{P}_{s,a,s'}^{(k)}\gamma}{\max\{N_{k,s,a}, 1\}} |\delta_{k,s,s'} - \delta_{s,s'}^*|}\right)$$

$$= O\left(\sqrt{H} \sum_{s,a} v_{k,s,a} \sum_{s'} \sqrt{\frac{\hat{P}_{s,a,s'}^{(k)}\gamma |\delta_{k,s,s'} - \delta_{s,s'}^*|}{\max\{N_{k,s,a}, 1\}}}\right).$$

$$(14)$$

To bound (13, according to (4) and the fact $v_{k,s,a} \leq \max\{N_{k,s,a}, 1\}$ we have $v_{k,s,a}\sqrt{\frac{\hat{P}_{s,a,s'}^{(k)}|\delta_{k,s,s'} - \delta_{s,s'}^*|}{\max\{N_{k,s,a}, 1\}}} \leq \sqrt{\max\{N_{k,s,a}, 1\}\hat{P}_{s,a,s'}^{(k)}|\delta_{k,s,s'} - \delta_{s,s'}^*|} = \tilde{O}(T^{\frac{1}{4}})$. To be rigorous, we have following lemma.

**Lemma 5.** *With probability $1 - S^2 T \delta$, it holds that*

$$\sum_k \sum_{s,a} v_{k,s,a} \sum_{s'} \sqrt{\frac{\hat{P}_{s,a,s'}^{(k)}|(\delta_{k,s,s'} - \delta_{s,s'}^*)|}{\max\{N_{k,s,a}, 1\}}} \leq 11KS^{\frac{5}{2}}A^{\frac{1}{4}}H^{\frac{1}{2}}T^{\frac{1}{4}}\gamma^{\frac{1}{4}}. \qquad (15)$$

Due to the lack of space, the proofs are delayed to the appendix.

Putting (9)-(12), (14), Lemma 3, Lemma 4 and Lemma 5 together, we conclude that $\mathcal{R}(T) = \tilde{O}(\sqrt{SAHT})$.

# 6 Conclusion

In this paper we answer the open problems proposed by Jiang and Agarwal [2018] partly by designing an OFU based algorithm EBF and proving a regret bound of $\tilde{O}(\sqrt{HSAT})$ whenever $H$, an upper bound on $sp(h^*)$ is known. We evaluate state-pair difference of the optimal bias function during learning process. Based on this evaluation, we design a delicate confidence set to guide the agent to explore in the right direction. We also prove a regret bound of $\tilde{O}(\sqrt{DSAT})$ without prior knowledge about $sp(h^*)$. Both two regret bounds match the corresponding lower bound up to a logarithmic factor and outperform the best previous known bound by an $\sqrt{S}$ factor.

## Acknowledgments

The authors would like to thank the anonymous reviewers for valuable comments and advice.

## Footnotes

[1] The symbol $\tilde{O}$ means $O$ with log factors ignored.

[2]In this paper, we use $\Delta^X$ to denote all distributions on $X$. Particularly, we use $\Delta^m$ to denote the *m-simplex*.

[3]In this paper, we use $[v_1, v_2, ..., v_S]^T$ to indicate a vector $v \in \mathbb{R}^{\mathcal{S}}$

[4]To explain the high-level idea, we assume this expectaion is well-defined.

[5]In this paper $\gamma$ always denotes $\log(\frac{2}{\delta})$.

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
