[Supplementary Material]

# Appendices

**Organization.** In Section A, we analysis the issues in the proof of [Agrawal & Jia, 2017]. In Section B, we give some basic lemmas (mainly concentration inequalities). Section C is devoted to the missing proofs in the analysis of Theorem 1. At last, we present the proof of Corollary 1 in Section D.

## A  Mistake in the Analysis of Previous Work

In this section we mainly analysis the mistake in the proof of Lemma C.2 and Lemma C.1 [Agrawal & Jia, 2017]. The lemma can be described as

**Lemma 6** (Lemma C.2, Agrawal & Jia, 2017). *Let $\hat{p}$ be the average of $n$ independent multinoulli trials with parameter $p \in \Delta^S$. Let*

$$Z := \max_{v \in [0,D]^S} (\hat{p} - p)^T v.$$

*Then $Z \leq D\sqrt{\frac{2\log(1/\rho)}{n}}$, with probability $1 - \rho$.*

We give a counter example as following. Suppose $D = 2$, $p_i = \frac{1}{S}$ for each $1 \leq i \leq S$, then we have $Z = \max_{v \in [0,2]^S} (\hat{p} - p)^T v = \max_{v \in [0,2]^S} (\hat{p} - p)^T (v - \mathbf{1}) = \max_{v \in [-1,1]^S} (\hat{p} - p)^T v = \sum_{i=1}^S |\hat{p}_i - \frac{1}{S}|$, and $\mathbb{E}[Z] = \sum_{i=1}^S \mathbb{E}[|\hat{p}_i - \frac{1}{S}|] = S\mathbb{E}[|\hat{p}_1 - \frac{1}{S}|]$ due to symmetry of $p$. Therefore, $\mathbb{E}[Z] = S\mathbb{E}[|\hat{p}_1 - \frac{1}{S}|] \geq (1 - \frac{1}{S})^n$. On the other hand, if Lemma 6 is right, by setting $\rho = \frac{1}{n}$ we have $\mathbb{E}[Z] \leq \sqrt{\frac{2\log(n)}{n}} + \frac{1}{n}$. Letting $S \to \infty$, it follows that $1 = \lim_{S \to \infty} (1 - \frac{1}{S})^n \leq 2\sqrt{\frac{2\log(n)}{n}} + \frac{2}{n}$, which is wrong when $n \geq 30$.

**Lemma 7** (Lemma C.1 [Agrawal & Jia, 2017). *] Let $\tilde{p} \sim Dirichlet(m\bar{p})$. Let*

$$Z := \max_{v \in [0,D]^S} (\tilde{p} - \bar{p})^T v.$$

*Then, $Z \leq D\sqrt{\frac{2\log(2/\rho)}{m}}$, with probability $1 - \rho$.*

Again, to build a counter example, let $D = 2$, $\bar{p}_i = \frac{1}{S}$ for any $i$. $\mathbb{E}[Z] = S\mathbb{E}[|\tilde{p}_1 - \frac{1}{S}|] \geq \frac{1}{2}(\mathbb{P}(\tilde{p}_1 < \frac{1}{2S}) + \mathbb{P}(\tilde{p}_1 > \frac{3}{2S}))$. Note that $\tilde{p}_1 \sim Beta(\frac{m}{S}, m - \frac{m}{S})$. When $m > 1$ and $S > m$, the density function of $\tilde{p}_1$ is $\frac{x^{\frac{m}{S}-1}(1-x)^{m-\frac{m}{S}}}{B(\frac{m}{S}, m-\frac{m}{S})}$ for $x \in (0, 1)$, which is decreasing in $x$. Therefore, we have that $\mathbb{P}(\tilde{p}_1 < \frac{1}{2S}) \geq \frac{1}{2}\mathbb{P}(\frac{1}{2S} \leq \tilde{p}_1 \leq \frac{3}{2S}) = \frac{1}{2}(1 - (\mathbb{P}(\tilde{p}_1 < \frac{1}{2S}) + \mathbb{P}(\tilde{p}_1 > \frac{3}{2S})))$, and thus $\mathbb{P}(\tilde{p}_1 < \frac{1}{2S}) + \mathbb{P}(\tilde{p}_1 > \frac{3}{2S}) \geq \frac{1}{3}$. As a result, $\mathbb{E}[Z] \geq \frac{1}{6}$, which contradicts to Lemma 7. Moreover, we find that the mistake in their proof lies in the derivation

$$\mathbb{E}[DY - Z | Z = z : z \in \mathcal{E}_v] = \mathbb{E}[DY - D\mathbb{E}[Y_v - Z | Z = z : z \in \mathcal{E}_v]$$
$$= \mathbb{E}[DY_v - D\mathbb{E}[Y_v] - (\hat{p} - p)^T v | (\hat{p} - p)^T v]$$
$$= \mathbb{E}[DY_v - \hat{p}^T v | \hat{p}^T v] = 0$$

Actually, $\{Z = z : z \in \mathcal{E}_v\} \subsetneq \{Z = z : z = (\hat{p} - p)^T v\}$ because given the value of $Z = z$, it's still unknown that which $v$ is selected to maximize $(\hat{p} - p)^T v$. More rigorously, we have $\mathbb{E}[\mathbb{E}[DY_v - \hat{p}^T v | Z = z, z \in \mathcal{E}_v] | Z \in \mathcal{E}_v] = \mathbb{E}[DY_v - \hat{p}^T v | Z \in \mathcal{E}_v] = p^T v - \mathbb{E}[\hat{p}^T v | Z \in \mathcal{E}_v] < 0$, since $(\hat{p} - p)^T v > 0$ conditioning on $Z$ in $\mathcal{E}_v$ (except for $\hat{p} = p$). This contradicts to the analysis of Lemma C.2 in [Agrawal & Jia, 2017], which says that $\mathbb{E}[DY_v - \hat{p}^T v | Z = z, z \in \mathcal{E}_v] = 0$.

Therefore, the algorithm in [Agrawal & Jia, 2017] may not reach the regret bound of $\tilde{O}(D\sqrt{SAT})$.

## B  Some Basic Lemmas

In this section, we present some useful lemmas. Some of them are well known so that we omit the proof.

**Lemma 8** (Azuma's Inequality). *Suppose $\{X_k\}_{k=0,1,2,3,..}$ is a martingale and $|X_{k+1} - X_k| < c$. Then for all positive integers $N$ and all positive $t$,*

$$\mathbb{P}(|X_N - X_0| \geq t) \leq 2exp(\frac{-t^2}{2Nc^2}). \tag{16}$$

*Let $t = c\sqrt{2N\log(2/\delta)}$, then $\mathbb{P}(|X_N - X_0| \geq t) \leq \delta$.*

**Lemma 9** (Bernstein Inequality). *Let $\{X_k\}_{k \geq 1}$ be independent zero-mean random variables. Suppose that $|X_k| \leq M$ for all $k$. Then, for all positive $t$*

$$\mathbb{P}(|\sum_{k=1}^{n} X_k| \geq t) \leq 2exp(-\frac{t^2}{2(\sum_{k=1}^{n} E[X_k^2] + \frac{1}{3}Mt)}). \tag{17}$$

*Let $t = 2\sqrt{\sum_{k=1}^{n} \mathbb{E}[X_k^2]\log(2/\delta)} + 2M\log(2/\delta)$, then $\mathbb{P}(|\sum_{k=1}^{n} X_k| \geq t) \leq \delta$.*

**Lemma 10.** *Let $\hat{p}_n$ be the average of $n$ independent multinomial trials with parameter $p \in \Delta^m$. Then, for any fixed vector $u \in \mathbb{R}^m$, with probability $1 - \delta$, it holds that*

$$|(\hat{p}_n - p)^T u| \leq 2\sqrt{\frac{V(p,u)\gamma}{n}} + 2\frac{sp(u)\gamma}{n}.$$

*Proof.* Given $u \in \mathbb{R}^m$ and $p \in \Delta^m$, let $\{X_k\}_{k\geq 1}$ be i.i.d. random variable s.t. $\mathbb{P}(X_k = u_i - p^T u) = p_i, \forall k$. Because $E[X_k^2] = V(p,u)$ and $\frac{1}{n}\sum_{k=1}^{n} X_k = (\hat{p}_n - p)^T u$, according to Lemma 9 we get that

$$\mathbb{P}(|(\hat{p}_n - p)^T u| \geq 2\sqrt{\frac{V(p,u)\gamma}{n}} + 2\frac{sp(u)\gamma}{n}) \leq \delta.$$

$\square$

**Lemma 11** (Freedman (1975)). *Let $(M_n)_{n\geq 0}$ be a martingale such that $M_0 = 0$. Let $V_n = \sum_{k=1}^{n} \mathbb{E}[(M_k - M_{k-1})^2|\mathcal{F}_{k-1}]$ for $n \geq 0$, where $\mathcal{F}_k = \sigma(M_1, M_2, ..., M_k)$. Then, for any positive $x$ and for any positive $y$,*

$$\mathbb{P}(M_n \geq nx, V_n \leq ny) \leq exp(-\frac{nx^2}{2(y + \frac{1}{3}x)}). \tag{18}$$

**Lemma 12.** *Suppose $M$ is a flat MDP. Let $h$ and $\rho$ denote the optimal bias function and the optimal average reward respectively. We run $N$ steps under $M$ and get a trajectory $L$ of length $N$. Then we have, no matter which action is chosen in each step, for each $n \in [N]$, with probability $1 - \delta$, it holds that*

$$|\sum_{i=1}^{n}(r_i - \rho)| \leq (2\sqrt{n\gamma} + 1)sp(h). \tag{19}$$

*Moreover, suppose that the reward is bounded in $[0,1]$, $n \geq 4\gamma sp(h)^2$ and $sp(h) \geq 10$, then with probability $1 - 2\delta$ it holds that*

$$|\sum_{i=1}^{n}(r_i - \rho)| \leq 4\sqrt{n\gamma sp(h)} + sp(h). \tag{20}$$

*Proof.* Let $M_0 = h_{s_1}$ and $M_n - M_{n-1} = h_{s_{n+1}} - h_{s_n} + r_n - \rho$ for $n \geq 1$. Then $\{M_n - M_0\}_{n\geq 0}$ is a martingale martingale difference sequence since $\mathbb{E}[h_{s_{n+1}} - h_{s_n} + r_n - \rho|\mathcal{F}_{n-1}] = \sum_a \mathbb{P}(a_t = a)[E][h_{s_{n+1}} - h_{s_n} + r_n - \rho|\mathcal{F}_{n-1}, a_t = a] = \sum_a \mathbb{P}(a_t = a)(P_{s_n,a}^T h - h_{s_n} + r_{s_n,a} - \rho) = 0$. Because $|M_n - M_{n-1}| \leq \max_a |P_{s_n,a}^T h - h_{s_{n+1}}| \leq sp(h)$, $V_n \leq nsp(h)^2$. Plug $y = sp(h)^2$ and $x = \frac{2\sqrt{\gamma}sp(h)}{\sqrt{n}}$ into (18), then (19) follows easily. To prove (20), we need to provide a tighter bound for $V_n$. For $v \in \mathbb{R}^S$, we use $v^2$ to denote the vector $[v_1^2, v_2^2, ..., v_S^2]^T$. Because $V_n = \sum_{k=1}^{n} \mathbb{E}[(M_k - M_{k-1})^2|\mathcal{F}_{k-1}] = \sum_{k=1}^{n} P_{s_k,a_k}^T h^2 - (P_{s_k,a_k}^T h)^2$ and $P_{s_k,a_k}^T h - h_{s_k} = \rho - r_{s_k,a_k}$, we have that

$$V_n \leq \sum_{k=1}^{n}(P_{s_k,a_k}^T h^2 - h_{s_k}^2) + \sum_{k=1}^{n}(sp(h)|\rho - r_{s_k,a_k}| + (\rho - r_{s_k,a_k})^2).$$

By the assumption the reward is bounded in $[0,1]$, we have $\rho \in [0,1]$ and $|\rho - r_{s_k,a_k}| \le 1$. Let $X_n = \sum_{k=1}^{n}(P_{s_k,a_k}^T h^2 - h_{s_{k+1}}^2) = V_n + h_{s_{n+1}}^2 - h_{s_1}^2$ for $n \ge 1$ and $X_0 = 0$. It's clear $\{X_n\}_{n\ge 0}$ is a martingale difference sequence and $|X_k - X_{k-1}| \le sp(h)^2$. According to Lemma 8, we have that

$$P(|X_n| \ge \sqrt{2n\gamma} sp(h)^2) \le \delta$$

Then it follows that with probability $1 - \delta$, $|V_n| \le (\sqrt{2n\gamma} + 1)sp(h)^2 + n(2sp(h) + 1)$. When $n \ge 4\gamma sp(h)^2$ and $sp(h) \ge 10$, we get $|V_n| \le 4nsp(h)$. Again, plugging $x = \frac{4\sqrt{\gamma sp(h)}}{\sqrt{n}}$ and $y = 4sp(h)$ into (18), noticing that $n \ge 16\gamma sp(h)$, we conclude that, with probability $1 - 2\delta$, $|\sum_{i=1}^{n}(r_i - \rho)| \le 4\sqrt{n\gamma sp(h)} + sp(h)$. $\qquad \square$

We introduce a technical lemma which is actually an expansion of Lemma 19, [Jaksch et al., 2010].

**Lemma 13.** *Suppose $\{x_n\}_{n=1}^{N}$ is sequence of positive real number with $x_1 = 1$ and $x_n \le \sum_{i=1}^{n-1} x_i$ for $n = 2, 3, ..., N - 1$. Then we have, for any $0 < \alpha < 1$,*

$$x_1 + \sum_{n=2}^{N} x_n (\sum_{i=1}^{n-1} x_i)^{-\alpha} \le \frac{2^\alpha}{1-\alpha}(\sum_{n=1}^{N} x_n)^{1-\alpha}.$$

*Moreover, in the case $\alpha = 1$, we have*

$$x_1 + \sum_{n=2}^{N} x_n (\sum_{i=1}^{n-1} x_i)^{-1} \le 1 + 2\log(\sum_{n=1}^{N} x_n).$$

*Proof.* Let $S_n = \sum_{1 \le i \le n} x_i$ for $n \ge 1$, then it follows $2S_n \ge S_{n+1}$ for $n \in [N-1]$. By basic calculus, when $\alpha < 1$, for $n \ge 2$ we have

$$S_n^{1-\alpha} - S_{n-1}^{1-\alpha} \ge (1-\alpha)x_n S_n^{-\alpha} \ge \frac{1-\alpha}{2^\alpha} x_n S_{n-1}^{-\alpha}.$$

Note that $S_1^{1-\alpha} = 1$, we then have $x_1 + \sum_{n=2}^{N} x_n S_{n-1}^{-\alpha} \le 1 + \frac{2^\alpha}{1-\alpha}\sum_{n=2}^{N}(S_n^{1-\alpha} - S_{n-1}^{1-\alpha}) \le \frac{2^\alpha}{1-\alpha}S_N^{1-\alpha} + 1 - \frac{2^\alpha}{1-\alpha} \le \frac{2^\alpha}{1-\alpha}S_N^{1-\alpha}$.
In the case $\alpha = 1$, for $n \ge 2$ we have

$$\log(S_n) - \log(S_{n-1}) \ge \frac{x_n}{S_n} \ge \frac{x_n}{2S_{n-1}}.$$

Note that $\log(S_1) = 0$, we then have $x_1 + \sum_{n=2}^{N} x_n S_{n-1}^{-1} \le 1 + 2(\log(S_n - \log(S_1))) = 1 + 2\log(S_n)$. $\qquad \square$

Applying Lemma 13 to $\{v_{k,s,a}\}_{k\ge 1}$, we have that for any $0 < \alpha < 1$

$$\sum_k \frac{v_{k,s,a}}{\max\{N_{k,s,a}, 1\}^\alpha} \le \frac{2^\alpha}{1-\alpha}(N_{s,a}^{(T)})^{1-\alpha}$$

Combining this inequality and Jenson's inequality, we get that

$$\sum_{k,s,a} \frac{v_{k,s,a}}{\max\{N_{k,s,a}, 1\}^\alpha} \le \frac{2^\alpha}{1-\alpha}SA(\frac{T}{SA})^{1-\alpha} \tag{21}$$

In the case $\alpha = 1$, we also have

$$\sum_{k,s,a} \frac{v_{k,s,a}}{\max\{N_{k,s,a}, 1\}} \le SA + 2SA\log(\frac{T}{SA}) \tag{22}$$

With a slightly abuse of notations, we use $N_{k,s,a}$ to denote $\max\{N_{k,s,a}, 1\}$ in the rest of the paper for simplicity.

## C  Missing Proofs in the Analysis of Theorem 1

In this section, we present the proofs of Lemma 1-5 and give a detailed proof of Theorem 1.

## C.1 Proof of Lemma 1

Let $h \in \mathbb{R}^S$ and $\rho \in \mathbb{R}$ be fixed. We define a Markov process $X$ with state space $\mathcal{S}$. Let $\{\mathcal{F}_t\}_{t\geq 1}$ be the corresponding filtered algebra, i.e., $\mathcal{F}_t = \sigma(X_1, ..., X_t)$. Let $s_1$ be the initial state. For each state $s$, there are some actions and each action $a$ is equipped with a transition probability vector $p_{s,a}$ and a reward $r'_{s,a} = h_s + \rho - p_{s,a}^T h$. In the $t$-th step, there is a policy $\pi_t$. We select an action according to $\pi_t$, then execute it and reach the next state. We then have $\mathbb{P}[p_t = p_{s_t,a}, r'_t = r'_{s_t,a}] = \pi_{t,a}$, where $p_t$ is transition probability and $r'_t$ is the reward in current step.

Then it is clear $\{(s_t, s_{t+1}, r'_t)\}_{t=1}^n$ is measurable with respect to $\mathcal{F}_n$. For any two different states $s, s' \in \mathcal{S}$, given a trajectory $L = \{(s_t, s_{t+1}, r'_t)\}_{t=1}^n$, we define an indicator function $I_{s,s'}(L, t)$ as following:

If $t \geq n+1$, $I_{s,s'}(L, t) = 0$. Otherwise, let $U = \{i|s_i \in \{s, s'\}, 1 \leq i \leq t\}$. If $U$ is empty, $I_{s,s'}(L, t) = 0$; else $I_{s,s'}(L, t) = \mathbb{I}[s_{i^*} = s]$ where $i^*$ be the maximal element of $U$.

Let $L$ be the $N$-step trajectory of $X$ and $I_{s,s'}(t) = I_{s,s'}(L, t)$. Note that $I_{s,s'}(t)$ is a random variable, and it only depends on $\{s_u\}_{u=1}^t$, which is measurable with respect to $\mathcal{F}_{t-1}$. Let $W_t = \sum_{u=1}^t I_{s,s'}(u)(r_u - h_{s_u} + h_{s_{u+1}} - \rho)$, then we have $\mathbb{E}[W_1] = 0$ and $\mathbb{E}[W_t - W_{t-1}|\mathcal{F}_{t-1}] = 0$ for $t \geq 2$. It follows that $\{W_t\}_{t=1}^N$ is a martingale with respect to $\{\mathcal{F}_t\}_{t=1}^N$. Because $|W_t - W_{t-1}| = |I_{s,s'}(t)(r'_t - h_{s_t} + h_{s_{t+1}} - \rho^*)| \leq \max_a |I_{s,s'}(t)(h_{s_{t+1}} - p_{s_t,a}^T h)| \leq sp(h)$ and $|W_1| \leq sp(h)$, by (16), we have that, for any $n \leq N$,

$$\mathbb{P}(|W_n| \geq \sqrt{2N\gamma}sp(h) + sp(h)) \leq \delta.$$

Then it follows that, with probability $1 - N\delta$, for any $n \in [N]$,

$$|W_n| \leq \sqrt{2N\gamma}sp(h) + sp(h).$$

Recall the notations in Definition 4, $ts_1(\mathcal{L}) := \min\{\min\{t|s_t = s\}, N+2\}$,

$$te_k(\mathcal{L}) := \min\{\min\{t|s_t = s', t > ts_k(\mathcal{L})\}, N+2\}, k \geq 1,$$
$$ts_k(\mathcal{L}) := \min\{\min\{t|s_t = s, t > te_{k-1}(\mathcal{L})\}, N+2\}, k \geq 2.$$

and $c(s, s', \mathcal{L}) := \max\{k|te_k(\mathcal{L}) \leq N+1\}$. According to the definition of $I_{s,s'}(t)$, for any $c \in [c(s, s', \mathcal{L})]$, we have

$$W_{te_c(\mathcal{L})-1} = \sum_{u=1}^c \left( \sum_{ts_u(\mathcal{L}) \leq t \leq te_u(\mathcal{L})-1} (r'_t - \rho) + h_{s'} - h_s \right).$$

Given an algorithm $\mathcal{G}$, we can view $\mathcal{G}$ as a function which maps previous samples, policies and current state to a policy in current state, and we use $\mathcal{G}_t := \mathcal{G}(s_t, (s_u, \pi_u, a_u, r_u, s_{u+1})_{u=1}^{t-1})$ to denote this policy. By setting $h = h^*, \rho = \rho^*, p_{s,a} = P_{s,a}$ and $\pi_t = \mathcal{G}_t$, we have $r_{s,a} = h_s^* + \rho^* - p_{s,a}^T h^* = r'_{s,a}$, since $M$ is flat. It then follows that

$$W_{te_c(\mathcal{L})-1} = \sum_{u=1}^c \left( \sum_{ts_u(\mathcal{L}) \leq t \leq te_u(\mathcal{L})-1} (r_t - \rho^*) + h_{s'} - h_s \right).$$

As we proved before, with probability $1 - N\delta$, it holds that for any $1 \leq n \leq N$,

$$|W_n| \leq \sqrt{2N\gamma}sp(h) + sp(h).$$

Because $1 \leq ts_c(\mathcal{L}) \leq te_c(\mathcal{L}) - 1 \leq N$ for any $1 \leq c \leq c(s, s', \mathcal{L})$, Lemma 1 follows easily.

## C.2 Proof of Lemma 2

Recall the definition of bad events.

$$B_{1,k} := \left\{ \exists (s,a), s.t. |(P_{s,a} - \hat{P}_{s,a}^{(k)})^T h^*| > 2\sqrt{\frac{V(P_{s,a}, h^*\gamma)}{N_{k,s,a}}} + 2\frac{sp(h^*\gamma)}{N_{k,s,a}} \right\},$$

$$B_{2,k} = \left\{ \exists (s,a,s'), s.t. |\hat{P}_{s,a,s'}^{(k)} - P_{s,a,s'}| > 2\sqrt{\frac{\hat{P}_{s,a,s'}^{(k)}\gamma}{N_{k,s,a}}} + \frac{3\gamma}{N_{k,s,a}} + \frac{4\gamma^{\frac{3}{4}}}{N_{k,s,a}^{\frac{3}{4}}} \right\},$$

$$B_{3,k} = \left\{ |\sum_{1 \leq t < t_k} (\rho^* - r_{s_t, a_t})| > 26HS\sqrt{AT\gamma}, \sum_{k' < k} \sum_{s,a} v_{k',s,a} reg_{s,a} > 22HS\sqrt{AT\gamma} \right\}$$

$$B_{4,k} = \left\{ \{(\pi^*, P^*, h^*, \rho^*)|\pi^* \text{is a deterministic optimal policy}\} \cap \mathcal{M}_k = \varnothing \right\},$$

$B_k = B_{1,k} \cup B_{2,k} \cup B_{3,k} \cup B_{4,k}$ and $B = \cup_{1 \le k \le K+1} B_k$.

It's easy to see that for each $k$, $B_{1,k}$ and $B_{2,k}$ indicate the events where the concentration inequalities fail, and thus have a small probability. Suppose $B_{k'}^C$ occurs for each $k' < k$, we get that the regret before the $k$-th episode does not exceed $\tilde{O}(HS\sqrt{AT})$ with high probability based on the analysis of REGAL.C.

To show $\mathbb{P}(B_{4,k})$ is small, we prove that, conditioned on $\cap_{1 \le k' < k} B_{k'}^C$ occurs, with high probability, it holds that $h^* \in \mathcal{H}$. Let $\pi^*$ be a deterministic optimal policy. Note that if (5)-(7) holds for any $s, a, s'$ with $P'(\pi) = P$ where $P$ is the true transition model, we then have $(\pi^*, P, h^*, \rho^*) \in \mathcal{M}_k$, since (8) holds due to the optimality of $\pi^*$. Putting all together, we can bound $\mathbb{P}(B)$ up to $\tilde{O}(S^3 A^2 T)\delta$.

Note that $t_{K+1} - 1 = T$, then $B_{K+1}$ is also well defined. Firstly, for each $k$, according to Lemma 10, we have $\mathbb{P}(B_{1,k}) \le SA\delta$ directly.

To bound the probability of $B_{2,k}$, let $(s, a)$ be fixed. Defining $g(x) = [x, 1-x]^T$ for $x \in [0, 1]$. Then we have $|x_1 - x_2| = \frac{1}{2}|g(x_1) - g(x_2)|_1 = \frac{1}{2} \sup_{y \in \{-1,1\}^2} (g(x_1) - g(x_2))^T y$ for $x_1, x_2 \in [0, 1]$. It follows that $\mathbb{P}(|x_1 - x_2| \ge 2\epsilon) \le 4 \sup_{y \in \{-1,1\}^2} \mathbb{P}((g(x_1) - g(x_2))^T y \ge \epsilon)$. Noting that $V(g(x), y) \le 4x$ for each $y \in \{-1, 1\}^2$, according to Lemma 10 we have, for any $y \in \{-1, 1\}^2$

$$\mathbb{P}(|(g(\hat{P}_{s,a,s'}^{(k)}) - g(P_{s,a,s'}))^T y| \ge 2\sqrt{\frac{4P_{s,a,s'}\gamma}{N_{k,s,a}}} + \frac{2\gamma}{N_{k,s,a}}) \le \delta$$

which means that $\mathbb{P}(|\hat{P}_{s,a,s'}^{(k)} - P_{s,a,s'}| \ge 2\sqrt{\frac{P_{s,a,s'}\gamma}{N_{k,s,a}}} + \frac{\gamma}{N_{k,s,a}}) \le 4\delta$. Suppose that the event $\{|\hat{P}_{s,a,s'}^{(k)} - P_{s,a,s'}| < 2\sqrt{\frac{P_{s,a,s'}\gamma}{N_{k,s,a}}} + \frac{\gamma}{N_{k,s,a}}\}$ occurs, then we have

$$
\begin{aligned}
|\hat{P}_{s,a,s'}^{(k)} - P_{s,a,s'}| &\le 2\sqrt{\frac{P_{s,a,s'}\gamma}{N_{k,s,a}}} + \frac{\gamma}{N_{k,s,a}} \\
&\le 2\sqrt{\frac{(\hat{P}_{s,a,s'}^{(k)} + 2\sqrt{\frac{\gamma}{N_{k,s,a}}} + \frac{\gamma}{N_{k,s,a}})\gamma}{N_{k,s,a}}} + \frac{\gamma}{N_{k,s,a}} \\
&\le 2\sqrt{\frac{\hat{P}_{s,a,s'}^{(k)}\gamma}{N_{k,s,a}} + \frac{3\gamma}{N_{k,s,a}} + \frac{4\gamma^{\frac{3}{4}}}{N_{k,s,a}^{\frac{3}{4}}}}.
\end{aligned}
$$

Therefore, $\mathbb{P}(B_{2,k}) \le 4S^2 A\delta$.

For $k = 1$, $B_{3,k}^C$ and $B_{4,k}^C$ holds trivially. For $k > 1$, assuming $\cap_{k' \ge 1} B_{1,k'}^C$, $\cap_{k' \ge 1} B_{2,k'}^C$, $\cap_{1 \le k' < k} B_{3,k'}^C$ and $\cap_{1 \le k' < k} B_{4,k'}^C$ hold. We start to bound $\mathbb{P}(B_{4,k})$. Note that $B_{3,k-1}^C$ ensures that

$$\sum_{1 \le k' < k} \sum_{s,a} v_{k,s,a} reg_{s,a} \le 22HS\sqrt{AT\gamma} \tag{23}$$

Note that if we replace the reward function $r_{s,a}$ by $r'_{s,a} = r_{s,a} + reg_{s,a}$, the MDP $M$ will be *flat*. According to Lemma 1, we have

$$\left| \sum_{i=1}^{c(s,s',\mathcal{L}_{t_k-1})} \sum_{ts_i \le j \le te_i - 1} (r_{s_j,a_j} + reg_{s_j,a_j} - \rho^*) - c(s, s', \mathcal{L}_{t_k-1})\delta^*_{s,s'} \right| \le (\sqrt{2T\gamma} + 1)H \tag{24}$$

with probability $1 - T\delta$. Combining (23) and (24), we get that

$$\left| \sum_{i=1}^{c(s,s',\mathcal{L}_{t_k-1})} \sum_{ts_i \le j \le te_i - 1} (r_{s_j,a_j} - \rho^*) - c(s, s', \mathcal{L}_{t_k-1})\delta^*_{s,s'} \right| \le (\sqrt{2T\gamma} + 1)H + 22HS\sqrt{AT\gamma} \tag{25}$$

Furthermore, $B_{3,k}^C$ also implies that $|\sum_{1 \le k' < k} \sum_{s,a} v_{k,s,a}(\rho^* - r_{s,a})| \le 26HS\sqrt{AT\gamma}$, then it follows $(\sum_{1 \le k' < k} l_{k'})|\hat{\rho}_k - \rho^*| \le 26HS\sqrt{AT\gamma}$ where $l_{k'}$ is the length of the $k'$-th episode and

$\hat{\rho}_k = \frac{\sum_{1 \le t \le t_k - 1} r_t}{\max\{\sum_{1 \le k' \le k} l_{k'}, 1\}}$ is the average reward before the $k$-th episode. Therefore, we have that

$$
\begin{aligned}
&| \sum_{i=1}^{c(s, s', \mathcal{L}_{t_k-1})} \sum_{ts_i \le j \le te_i - 1} (r_{s_j, a_j} - \hat{\rho}_k) - c(s, s', \mathcal{L}_{t_k-1}) \delta_{s,s'}^* | \\
&\le | \sum_{i=1}^{c(s, s', \mathcal{L}_{t_k-1})} \sum_{ts_i \le j \le te_i - 1} (r_{s_j, a_j} - \rho^*) - c(s, s', \mathcal{L}_{t_k-1}) \delta_{s,s'}^* | + |(\sum_{1 \le k' < k} l_{k'})(\hat{\rho}_k - \rho^*)| \\
&\le (\sqrt{2T\gamma} + 1)H + 48HS\sqrt{AT\gamma}
\end{aligned}
\tag{26}
$$

which means that $h^* \in \mathcal{H}$ in the beginning of the $k$-th episode.

The last step is to prove that (5), (6) and (7) hold for $P'(\pi) = P$ with high probability. (5) holds evidently because of $B_{2,k}^C$. According to the $L_1$ norm concentration inequality [Weissman et al,. 2003], we see that $\mathbb{P}(|P_{s,a} - \hat{P}_{s,a}^{(k)}| \le \sqrt{\frac{12S\gamma}{N_{k,s,a}}}) \le \delta$, thus (6) is satisfied. In order to prove (7) holds for $P' = P$ with high probability, by using Lemma 10 twice, we have that for each $(s, a)$

$$
\begin{aligned}
|(P_{s,a} - \hat{P}_{s,a}^{(k)})^T h^*| &\le 2\sqrt{\frac{V(P_{s,a}, h^*)\gamma}{N_{k,s,a}}} + 2\frac{H\gamma}{N_{k,s,a}} \\
&\le 2\sqrt{\frac{V(\hat{P}_{s,a}^{(k)}, h^*)\gamma}{N_{k,s,a}}} + 2\sqrt{\frac{|V(P_{s,a}, h^*) - V(\hat{P}_{s,a}^{(k)}, h^*)|\gamma}{N_{k,s,a}}} + 2\frac{H\gamma}{N_{k,s,a}} \\
&\le 2\sqrt{\frac{V(\hat{P}_{s,a}^{(k)}, h^*)\gamma}{N_{k,s,a}}} + 2\sqrt{\frac{H^2(2\sqrt{\frac{\gamma}{N_{k,s,a}}} + 2\frac{\gamma}{N_{k,s,a}})\gamma}{N_{k,s,a}}} + 2\frac{H\gamma}{N_{k,s,a}} \\
&\le 2\sqrt{\frac{V(\hat{P}_{s,a}^{(k)}, h^*)\gamma}{N_{k,s,a}}} + 12\frac{H\gamma}{N_{k,s,a}} + 10\frac{H\gamma^{3/4}}{N_{k,s,a}^{3/4}}.
\end{aligned}
$$

holds with probability $1 - 2\delta$. Therefore, $\mathbb{P}(B_{4,k}^C) \le (T + 3SA)\delta$.

On the other side, note that $\cap_{1 \le k' < k} B_{4,k'}^C$ ensures that $\{(\pi^*, P^*, h^*, \rho^*) | \pi^* \in \mathcal{O}\} \cap \mathcal{M}_k \ne \varnothing$. It means that $\rho(\pi_k) \ge \rho^*$. Following the proof of Theorem 2 [Bartlett and Tewari, 2009], we get that when $T \ge A\log(T)$

$$
\begin{aligned}
\sum_{1 \le t \le t_k - 1} (\rho^* - r_t) &\le |\sum_k v_k^T (P_k' - P_k)|_1 H + |\sum_k v_k^T (P_k - I) h_k| \\
&\le 2H(\sum_{k,s,a} v_{k,s,a} \sqrt{\frac{12S\gamma}{N_{k,s,a}}} + \sqrt{2T\gamma} + K) \\
&\le 18HS\sqrt{AT\gamma}
\end{aligned}
$$

with probability $1 - 2AT\delta$. Moreover, note that

$$
\sum_{1 \le t \le t_k - 1} reg_{s_t, a_t} = \sum_{1 \le t \le t_k - 1} (\rho^* - r_t) + \sum_{1 \le t \le t_k - 1} (h_{s_t}^* - P_{s_t, a_t}^T h^*)
\tag{27}
$$

By Azuma's inequality (Lemma 8), we have that

$$
|\sum_{1 \le i \le t} (h_{s_i}^* - P_{s_i, a_i}^T h^*)| \le 2H + \sqrt{2T\gamma}H
\tag{28}
$$

holds for any $1 \le t \le T$ with probability $1 - T\delta$. Assuming (27) and (28) hold for any $1 \le t \le T$, noticing that $reg_{s,a} \ge 0$ for any $(s, a)$, we have

$$
|\sum_{1 \le t \le t_k - 1} reg_{s_t, a_t}| \le 18HS\sqrt{AT\gamma} + 2H + \sqrt{2T\gamma}H \le 22HS\sqrt{AT\gamma}
$$

and

$$| \sum_{1 \leq t \leq t_k - 1} (\rho^* - r_t)| \leq | \sum_{1 \leq t \leq t_k - 1} reg_{s_t, a_t}| + | \sum_{1 \leq i \leq t} (h^*_{s_i} - P^T_{s_i, a_i} h^*)| \leq 26 H S \sqrt{AT\gamma}$$

At last, we conclude that when $\cap_{k' \geq 1} B^C_{1,k'}$, $\cap_{k' \geq 1} B^C_{2,k'}$, $\cap_{1 \leq k' < k} B^C_{3,k'}$ and $\cap_{1 \leq k' < k} B^C_{4,k'}$ hold, $\mathbb{P}(B_{3,k}) \leq (2AT + T)\delta$.

Putting all together we have

$$\mathbb{P}(B) \leq (K+1)(2AT + 8S^2 A + 2T)\delta \leq (6AT + 12S^2 A)SA \log(T)\delta$$

when $T \geq A \log(T)$ and $SA \geq 4$.

### C.3 Proof of Lemma 3

**Lemma 14.** *Let* $V = \sum_k \sum_{s,a} v_{k,s,a} V(P_{s,a}, h_k)$ *and* $W = \sum_k ①_k$. *For any* $C > 0$, *we have*

$$\mathbb{P}(|V| \leq C, |W| \geq KH + (4H + 2\sqrt{C})\gamma) \leq 2\delta$$

*Proof.* Let $X_{k,n} = \sum_{i=1}^{n} (P^T_{s_{k,i}, a_{k,i}} h_k - h_{k, s_{k,i+1}})$ where $(s_{k_i}, a_{k_i}, r_{k_i}, s_{k_{i+1}})$ is the $i$-th sample in the $k$-th episode. We use $l_k$ to denote the length of the $k$-th episode. Let $e_n = \max\{k | t_k \leq n\}$ and $Z_n = \sum_{k=1}^{e_n - 1} X_{k, l_k} + X_{e_n, n - t_{e_n} + 1}$. Let $\mathcal{F}_n = \sigma(Z_1, ..., Z_n)$. It's easy to see $E[Z_{n+1} - Z_n | \mathcal{F}_n] = E[X_{e_n, n+2 - t_{e_n}} - X_{e_n, n+1 - t_{e_n}} | \mathcal{F}_n] = 0$ if $e_n = e_{n+1}$, and $E[Z_{n+1} - Z_n | \mathcal{F}_n] = E[X_{e_{n+1}, 1} | \mathcal{F}_n] = 0$ otherwise. Therefore, $\{Z_n\}_{n \geq 1}$ is a martingale with respect to $\{\mathcal{F}_n\}_{n \geq 1}$. On the other hand, it's easy to see $|Z_{n+1} - Z_n| \leq H$, We then apply Lemma 11 to $\{Z_n\}_{n \geq 1}$ with $n = T$, $nx = (2\sqrt{C} + 4H)\gamma$ and $ny = C$, and obtain that

$$\mathbb{P}(Z_T \geq 2\sqrt{C}\gamma + 4H\gamma, |V| \leq C) \leq \delta$$

At last, because $|W - Z_T| = |\sum_k -h_{k,s_1} + h_{k, s_{l_k+1}}| \leq KH$, we conclude that,

$$\mathbb{P}(|V| \leq C, |W| \geq KH + (4H + 2\sqrt{C})\gamma) \leq 2\delta.$$

$\square$

Note that $①_k = v_k^T (P_k - I)^T h_k = \sum_{i=1}^{n} (P^T_{s_i, a_i} h_k - h_{k, s_i}) = \sum_{i=1}^{l_k} (P^T_{s_i, a_i} h_k - h_{k, s_{i+1}}) - h_{k, s_1} + h_{k, s_{l_k+1}}$. Let $X_n = \sum_{i=1}^{n} (P^T_{s_i, a_i} h_k - h_{k, s_{i+1}})$. Now it suffices to show that $\sum_k \sum_{s,a} v_{k,s,a} V(P_{s,a}, h_k) = O(TH)$ w.h.p.. Let $x^2$ denote the vector $[x_1^2, ..., x_S^2]^T$ for $x = [x_1, ..., x_S]^T$. Note that

$$\sum_k \sum_{s,a} v_{k,s,a} V(P_{s,a}, h_k) = \sum_k \sum_{s,a} v_{k,s,a} (P^T_{s,a} h_k^2 - ((P'_{k,s,a})^T h_k)^2)$$
$$+ \sum_k \sum_{s,a} v_{k,s,a} (P'_{k,s,a} - P_{s,a})^T h_k (P'_{k,s,a} + P_{s,a})^T h_k. \tag{29}$$

By the definition of $h_k$, we have that $(P'_{k,s,a})^T h_k - h_{k,s} = \rho_k - r_{s,a}$. Then we obtain that,

$$| \sum_{k,s,a} v_{k,s,a} (P^T_{s,a} h_k^2 - ((P'_{k,s,a})^T h_k)^2)| = | \sum_{k,s,a} v_{k,s,a} (P^T_{s,a} h_k^2) - h^2_{k,s}| + | \sum_{k,s,a} h^2_{k,s} - (h_{k,s} + \rho_k - r_{s,a})^2|$$
$$\leq | \sum_{k,s,a} v_{k,s,a} (P^T_{s,a} h_k^2) - h^2_{k,s}| + | \sum_{k,s,a} (\rho_k - r_{s,a})(2h_{k,s} + \rho_k - r_{s,a})|$$
$$\leq \sum_{k,s,a} v_{k,s,a} (P^T_{s,a} h_k^2 - h^2_{k,s}) + \sum_{k,s,a} v_{k,s,a} (2H + 1) \tag{30}$$

According to Lemma (8), we have that, with probability $1 - \delta$

$$\sum_{k,s,a} v_{k,s,a} (P^T_{s,a} h_k^2 - h^2_{k,s}) \leq \sqrt{2T\gamma} H^2 + KH^2 \tag{31}$$

Combining (30) and (31), we have that, with probability $1 - \delta$, it holds that

$$|\sum_{k,s,a} v_{k,s,a}(P_{s,a}^T h_k^2 - ((P_{k,s,a}')^T h_k)^2)| \leq \sqrt{2T\gamma}H^2 + KH^2 + T(2H + 1) \qquad (32)$$

Assuming the good event $G$ occurs, the second term in (29) can be bounded by $4H^2 \sum_{k,s,a} v_{k,s,a}\sqrt{\frac{S\gamma}{N_{k,s,a}}}$. Combining this with (32), we obtain that, with probability $1 - \delta$, it holds that

$$\sum_k \sum_{s,a} v_{k,s,a}V(P_{s,a}, h_k) \leq \sqrt{2T\gamma}H^2 + KH^2 + T(2H + 1)) + 4\sqrt{2}H^2 S\sqrt{AT\gamma} \qquad (33)$$

The dominant term is the right hand side of (33) is $2TH$ when $T$ is large enough. Specifically, when $T \geq S^2AH^2\gamma$, we have $\sum_k \sum_{s,a} v_{k,s,a}V(P_{s,a}, h_k) \leq 12TH$.

Let $C = 12TH$ in Lemma 14, then it follows that

$$\mathbb{P}(|\sum_k \text{①}_k| \geq KH + (4H + 2\sqrt{12TH})\gamma \leq \mathbb{P}(\sum_k \sum_{s,a} v_{k,s,a}V(P_{s,a}, h_k) \geq 12TH) +$$

$$\mathbb{P}(\sum_k \sum_{s,a} v_{k,s,a}V(P_{s,a}, h_k) \leq 12TH, |\sum_k \text{①}_k| \geq KH + (4H + 2\sqrt{12TH})\gamma)$$

$$\leq 3\delta.$$

## C.4 Proof of Lemma 4

**Lemma 15.** *When $T \geq H^2S^2A\gamma$, with probability $1 - \delta$, it holds that $\sum_{s,a} N_{s,a}^{(T)}V(P_{s,a}, h^*) \leq 49TH$*

*Proof.* Noting that $P_{s,a}^T h^* = h_s^* + \rho^* - r_{s,a} - reg_{s,a}$, we have

$$\sum_{s,a} N_{s,a}^{(T)}V(P_{s,a}, h^*) = \sum_{s,a} N_{s,a}^{(T)}(P_{s,a}^T h^{*2} - (P_{s,a}^T h^*)^2)$$

$$= \sum_{s,a} N_{s,a}^{(T)}(P_{s,a}^T h^{*2} - h_s^{*2}) + \sum_{s,a} N_{s,a}^{(T)}(reg_{s,a} + r_{s,a} - \rho^*)(P_{s,a}^T h^* + h_s^*)$$

$$\leq \sqrt{2T\gamma}H^2 + KH^2 + 2H\sum_{s,a} N_{s,a}^{(T)}reg_{s,a} + 2TH$$

$$\qquad (34)$$

with probability $1 - \delta$. By definition of $B_{3,K+1}^C$, we have $\sum_{s,a} N_{s,a}^{(T)}reg_{s,a} \leq 22HS\sqrt{AT\gamma}$. By combining this inequality with (34), when $T \geq H^2S^2A\gamma$, we have

$$\sum_{s,a} N_{s,a}^{(T)}V(P_{s,a}, h^*) \leq 2TH + H^2(44S\sqrt{AT\gamma} + \sqrt{2T\gamma} + K) \leq 49TH$$

holds with probability $1 - \delta$. $\qquad \square$

Assuming (34) holds, we have that

$$\sum_{k,s,a} v_{k,s,a}\sqrt{\frac{V(P_{s,a}, h^*)\gamma}{N_{k,s,a}}} = \sum_{s,a} \sqrt{V(P_{s,a}, h^*)\gamma}\sum_k v_{k,s,a}\sqrt{\frac{1}{N_{k,s,a}}}$$

$$\leq 2\sqrt{2}\sum_{s,a} \sqrt{N_{s,a}^{(T)}V(P_{s,a}, h^*)\gamma}$$

$$\qquad (35)$$

$$\leq 2\sqrt{2SA\gamma}\sqrt{\sum_{s,a} N_{s,a}^{(T)}V(P_{s,a}, h^*)}$$

$$\leq 21\sqrt{SAHT\gamma}.$$

Here the first inequality is by Lemma 13 with $\alpha = \frac{1}{2}$, the second inequality is Jenson's inequality and (34) implies the last inequality. Obviously, Lemma 4 follows by Lemma 15.

## C.5 Proof of Lemma 5

Note that if we replace the reward $r_{s,a}$ by $r_{s,a} + reg_{s,a}$, then the MDP $M$ would be a *flat* MDP. According to Lemma 1, we have that, with probability $1 - S^2 T \delta$, for any $t \leq T$ and two different states $s, s'$, it holds that

$$| \sum_{k=1}^{c(s,s',\mathcal{L}_{t_k})} \sum_{ts_k \leq i \leq te_k(\mathcal{L})-1} (r_i + reg_{s_i,a_i} - \rho^*) - c(s,s',\mathcal{L}_{t_k})\delta^*_{s,s'}| \leq (\sqrt{2T\gamma}+1)H$$

At the same time, $B^C_{4,k}$ implies (26) is true for $t = t_k$. Then we have

$$| \sum_{k=1}^{c(s,s',\mathcal{L}_{t_k})} \sum_{ts_k \leq i \leq te_k(\mathcal{L})-1} (r_i - \hat{\rho}_k) - c(s,s',\mathcal{L}_{t_k})\delta_{k,s,s'}| \leq (\sqrt{2T\gamma}+1)H + 48HS\sqrt{AT\gamma}$$

Because $B^C_{3,k}$ occurs, $(t_k - 1)|\rho^* - \hat{\rho}_k| \leq 26HS\sqrt{AT\gamma}$ and $\sum_{1 \leq k' < k} reg_{s_{k'},a_{k'}} \leq 22HS\sqrt{AT\gamma}$. Let $N_{k,s,a,s'} = \sum_{1 \leq t \leq t_k-1} I[s_t = s, a_t = a, s_{t+1} = s']$. Because $|a-b| \leq |a+c|+|b+d|+|c|+|d|$, by letting

$$a = \sum_{k=1}^{c(s,s',\mathcal{L}_{t_k})} \sum_{ts_k \leq i \leq te_k(\mathcal{L})-1} (r_i - \rho^*) - c(s,s',\mathcal{L}_{t_k})\delta^*_{s,s'},$$

$$b = \sum_{k=1}^{c(s,s',\mathcal{L}_{t_k})} \sum_{ts_k \leq i \leq te_k(\mathcal{L})-1} (r_i - \rho^*) - c(s,s',\mathcal{L}_{t_k})\delta_{k,s,s'},$$

$$c = \sum_{k=1}^{c(s,s',\mathcal{L}_{t_k})} \sum_{ts_k \leq i \leq te_k(\mathcal{L})-1} reg_{s_i,a_i}, \quad d = \sum_{k=1}^{c(s,s',\mathcal{L}_{t_k})} \sum_{ts_k \leq i \leq te_k(\mathcal{L})-1} (\rho^* - \hat{\rho}_k),$$

we have that

$$|N_{k,s,a,s'}(\delta_{k,s,s'} - \delta^*_{s,s'})| \leq |c(s,s',\mathcal{L}_{t_k})(\delta_{k,s,s'} - \delta^*_{s,s'})| \leq 2(\sqrt{2T\gamma}+1)H + 96HS\sqrt{AT\gamma}$$

and

$$\sum_k \sum_{s,a} v_{k,s,a} \sum_{s'} \sqrt{\frac{\hat{P}^{(k)}_{s,a,s'}|(\delta_{k,s,s'} - \delta^*_{s,s'})|}{N_{k,s,a}}}$$

$$= \sum_{k,s,a} \frac{v_{k,s,a}}{N_{k,s,a}} \sum_{s'} \sqrt{N_{k,s,a,s'}|(\delta_{k,s,s'} - \delta^*_{s,s'})|} \quad (36)$$

$$\leq KS^2 \sqrt{2(\sqrt{2T\gamma}+1)H + 96HS\sqrt{AT\gamma}}$$

$$\leq 11KS^{\frac{5}{2}}A^{\frac{1}{4}}H^{\frac{1}{2}}T^{\frac{1}{4}}\gamma^{\frac{1}{4}},$$

where the first inequality holds because $\sum_{k,s,a} \frac{v_{k,s,a}}{N_{k,s,a}} \leq \sum_{k,s,a} \mathbb{I}[\pi_k(s) = a] \leq KS$.

## C.6 Detailed Proof of Theorem 1

According to Lemma 2, the probability of bad event is bounded by $(6AT + 12S^2A)SA\log(T)$ when $T \geq A\log(T)$ and $SA \geq 4$. We then consider to bound the regret when the good event occurs. We present more rigorous analysis compared to the proof sketch in Section 5.2. Recall that

$$\mathcal{R}_k = v_k^T(\rho^*\mathbf{1} - r_k) \leq v_k^T(\rho_k\mathbf{1} - r_k) = v_k^T(P_k' - I)^T h_k$$
$$= \underbrace{v_k^T(P_k - I)^T h_k}_{①_k} + \underbrace{v_k^T(\hat{P}_k - P_k)^T h^*}_{②_k} + \underbrace{v_k^T(P_k' - \hat{P}_k)^T h_k}_{③_k} + \underbrace{v_k^T(\hat{P}_k - P_k)^T(h_k - h^*)}_{④_k};$$

$$②_k \leq \sum_{s,a} v_{k,s,a}\left(2\sqrt{\frac{V(P_{s,a}, h^*)\gamma}{N_{k,s,a}}} + 2\frac{H\gamma}{N_{k,s,a}}\right), \quad (37)$$

$$\sqrt{V(\hat{P}_{s,a}^{(k)}, h_k)} - \sqrt{V(P_{s,a}, h^*)} \le \sum_{s'} \sqrt{4H\hat{P}_{s,a,s'}^{(k)}|\delta_{k,s,s'} - \delta_{s,s'}^*|} + \sqrt{4H^2\sqrt{\frac{14S\gamma}{N_{k,s,a}}}}. \qquad (38)$$

Plugging (38) into (11), we get that

$$③_k \le \sum_{s,a} v_{k,s,a} L_2(N_{k,s,a}, \hat{P}_{s,a}^{(k)}, h_k) = \sum_{s,a} v_{k,s,a}\left(2\sqrt{\frac{V(\hat{P}_{s,a}^{(k)}, h_k)\gamma}{N_{k,s,a}}} + 12\frac{H\gamma}{N_{k,s,a}} + 10\frac{H\gamma^{3/4}}{N_{k,s,a}^{3/4}}\right)$$

$$\le \sum_{s,a} v_{k,s,a}\left(2\sqrt{\frac{V(P_{s,a}, h^*)\gamma}{N_{k,s,a}}} + 4\sum_{s'}\sqrt{\frac{H\hat{P}_{s,a,s'}^{(k)}|\delta_{k,s,s'} - \delta_{s,s'}^*|\gamma}{N_{k,s,a}}} + 8\frac{HS^{\frac{1}{4}}\gamma^{3/4}}{N_{k,s,a}^{3/4}} + 12\frac{H\gamma}{N_{k,s,a}} + 10\frac{H\gamma^{3/4}}{N_{k,s,a}^{3/4}}\right).$$

$$(39)$$

Based on (14), $B_{2,k}^C$ and the fact $|\delta_{k,s,s'} - \delta_{s,s'}^*| \le 2H$, we have that

$$④_k = \sum_{s,a} v_{k,s,a}(\hat{P}_{s,a}^{(k)} - P_{s,a})^T(h_k - h_{k,s}\mathbf{1} - h^* + h_s^*\mathbf{1}) = \sum_{s,a} v_{k,s,a}\sum_{s'}(\hat{P}_{s,a,s'}^{(k)} - P_{s,a,s})(\delta_{s,s'}^* - \delta_{k,s,s'})$$

$$\le \sum_{s,a} v_{k,s,a}\sum_{s'}(2\sqrt{\frac{\hat{P}_{s,a,s'}^{(k)}\gamma}{N_{k,s,a}}} + \frac{3\gamma}{N_{k,s,a}} + \frac{4\gamma^{3/4}}{N_{k,s,a}^{3/4}})|\delta_{k,s,s'} - \delta_{s,s'}^*|$$

$$\le 2\sum_{k,s,a} v_{k,s,a}\left(\sum_{s'}\sqrt{\frac{2H\hat{P}_{s,a,s'}^{(k)}|\delta_{k,s,s'} - \delta_{s,s'}^*|}{N_{k,s,a}}} + \frac{6SH\gamma}{N_{k,s,a}} + \frac{8SH\gamma^{3/4}}{N_{k,s,a}^{3/4}}\right)$$

$$(40)$$

Taking sum of RHS of (37), (39) and (40), based on the fact $S \ge 1$ we obtain that

$$②_k + ③_k + ④_k \le \sum_{s,a} v_{k,s,a}\left(4\sqrt{\frac{V(P_{s,a}, h^*)\gamma}{N_{k,s,a}}} + 20\frac{SH\gamma}{N_{k,s,a}} + 7\sum_{s'}\sqrt{\frac{H\hat{P}_{s,a,s'}^{(k)}|\delta_{k,s,s'} - \delta_{s,s'}^*|\gamma}{N_{k,s,a}}} + 26\frac{SH\gamma^{3/4}}{N_{k,s,a}^{3/4}}\right)$$

$$(41)$$

According to (9),(41) Lemma 4, Lemma 5 and Lemma 13, we obtain that when $T \ge S^3AH^2\gamma$ and $SA \ge 4$, with probability at least $1 - 20S^3A^2T\log(T)\delta$, it holds that

$$\mathcal{R}(T) = \sum_k \mathcal{R}_k \le KH + (4H + 2\sqrt{TH})\gamma$$

$$+ \sum_{k,s,a} v_{k,s,a}\left(4\sqrt{\frac{V(P_{s,a}, h^*)\gamma}{N_{k,s,a}}} + 20\frac{SH\gamma}{N_{k,s,a}} + 7\sum_{s'}\sqrt{\frac{H\hat{P}_{s,a,s'}^{(k)}|\delta_{k,s,s'} - \delta_{s,s'}^*|\gamma}{N_{k,s,a}}} + 26\frac{SH\gamma^{3/4}}{N_{k,s,a}^{3/4}}\right)$$

$$\le KH + (4H + 2\sqrt{TH})\gamma + 84\sqrt{SAHT\gamma} + 77KS^{\frac{5}{2}}A^{\frac{1}{4}}HT^{\frac{1}{4}}\gamma^{\frac{3}{4}}$$

$$+ 20SH\gamma(1 + 2SA\log(T)) + 208S^{\frac{7}{4}}A^{\frac{3}{4}}T^{\frac{1}{4}}H\gamma^{\frac{3}{4}} = \tilde{O}(\sqrt{SATH}).$$

$$(42)$$

Let $\delta_1 = 20S^3A^2T\log(T)\delta$. When $T \ge \{S^{12}A^3H^2, H^2SA\kappa, HSA\log(T)^2\kappa, H^2S^2\log(T)\kappa\}$ where $\kappa = \log(\frac{40S^3A^2T\log(T)}{\delta_1})$, with probability $1 - \delta_1$, we have that

$$\mathcal{R}(T) \le 490\sqrt{SATHlog(\frac{40S^2A^2Tlog(T)}{\delta_1})}.$$

**The selection of $p_1$:** Let $p_1(S, A, H, \log(\frac{1}{\delta})) = 64\log(\frac{1}{\delta}))^2(S^4A^4H^6 + S^4A^4H^4 + S^6A^2H^6) + S^{12}A^3H^3 + 100$. When $T \ge p_1(S, A, H, \log(\frac{1}{\delta}))$ and $S, A \ge 20$, we have that $T \ge S^{12}A^3H^3$ and $\frac{T}{\log^3(T)} \ge \sqrt{T} \ge 8\log(\frac{1}{\delta})\max\{S^2A^2H^3, S^3AH^3\} \ge \frac{1}{\log(T)}\max\{H^2SA\kappa, HSA\log(T)^2\kappa, H^2S^2\log(T)\kappa\}$, since $8SA \ge \frac{\kappa}{\log(\frac{1}{\delta})\log(T)}$. Therefore, $T \ge \max\{S^{12}A^3H^2, H^2SA\kappa, HSA\log(T)^2\kappa, H^2S^2\log(T)\kappa\}$.

## D    Proof of Corollary 1

In this section we consider to learn MDPs with finite diameter. According to Theorem 1, in order to reach an $\tilde{O}(\sqrt{DSAT})$ upper bound for the regret, it suffices to provide a real number $H$ such that $sp(h^*) \leq H \leq D$ within $o(\sqrt{T})$ steps. For a transition model $P$, we use $P^{(x,y)}$ to denote the transition model satisfying that $P_{s,a}^{(x,y)} = P_{s,a}$ when $s \neq x$, and $P_{s,a}^{(x,y)} = \mathbf{1}_y$ [6] when $s = x$, $\forall a$. Let $D_{xy} = \min_{\pi:\mathcal{S}\to\Delta_\mathcal{A}} T_{x\to y}^\pi$, then we try to learn $D_{xy}$ directly.

In Algorithm 3, when we start from $x$, we target to reach $y$ as soon as possible by employing a UCRL2-like algorithm. Once we reach $y$, we change the target to achieve $x$. Let $mdp(P, r)$ denote the MDP with transition model $P$ and reward function $r$. We maintain the two learning process separately, so they are corresponding to running two independent learning processes, which learn $mdp(P^{(y,x)}, \mathbf{1}_y)$ and $mdp(P^{(x,y)}, \mathbf{1}_x)$ respectively. Based on Algorithm 3, we can get a close approximation for $D_{xy}$ within $T^{\frac{1}{4}}$ steps. Without loss of generality, we assume $T^{\frac{1}{4}}$ is an integer.

**Lemma 16.** *When $T \geq (136D^3S\sqrt{A\gamma})^8$, for any $x \neq y \in \mathcal{S}$, let $(\hat{D}_{xy}, \hat{D}_{yx})$ be the output of Algorithm 3 with $(T^{1/4}, \delta, x, y)$ as the input, then with probability $1 - 8SAT^{\frac{1}{2}}\delta$, it holds that $|\hat{D}_{xy} - D_{xy}| \leq 1$ and $|\hat{D}_{yx} - D_{yx}| \leq 1$.*

*Proof of Corollary 1*. Obviously, an MDP with finite diameter is weak-communicating. We run Algorithm 3 for all $s \neq s'$ with $T_0 = T^{1/4}$ and $\delta_0 = \delta$ (without loss of generality, we assume that $T^{\frac{1}{4}}$ is an integer.). Denote the output of Algorithm 3 with input $(T^{1/4}, \delta, s, s')$ as $(\hat{D}_{ss'}, \hat{D}_{s's})$. Let $\hat{H} = \max_{s,s'} \hat{D}_{ss'} + 1$. According to Lemma 16, $sp(h^*) \leq \max_{s,s'} D_{ss'} \leq \hat{H} \leq D + 2$ with probability $1 - 8S^3AT^{\frac{1}{2}}\delta$. We then execute Algorithm 1 with $H = \hat{H}$ for $T - S(S-1)T^{\frac{1}{4}}$ steps. Since the total number of time steps for performing Algorithm 3 is at most $S^2T^{\frac{1}{4}}$, the regret in the first stage is at most $S^2T^{\frac{1}{4}}$. According to Theorem 1, when $T \geq 2\max\{(136D^3S\sqrt{A\kappa})^8, S^{12}A^3D^2, DSAlog^2(T)\kappa, D^2SA\kappa, D^2S^2log(T)\kappa\}$ where $\kappa = log(\frac{44S^2A^2Tlog(T)}{\delta_1})$, the regret can be bounded as

$$\mathcal{R}(T) \leq 491\sqrt{SATD(log(\frac{S^3A^2Tlog(T)}{\delta})}.$$

,with probability $1 - \delta$, the regret is at most $491\sqrt{SATD\log(\frac{44S^2A^2T\log(T)}{\delta_1})}$.     $\square$

**The selection of $p_2$:** Let $p_2(S, A, D, \log(\frac{1}{\delta})) = 4(136D^3S\sqrt{A})^{16}(8SA)^8 + log(\frac{1}{\delta})^8 10^{16}$. When $T \geq p_2(S, A, D, \log(\frac{1}{\delta}))$ and $S, A, D \geq 20$, $\frac{T}{\log(\frac{1}{\delta})^4\log(T)^4} \geq \sqrt{T} \geq 2(136D^3S\sqrt{A})^8(8SA)^4 \geq \frac{2(136D^3S\sqrt{A\kappa})^8}{\log(\frac{1}{\delta})^4\log(T)^4}$, since $8SA \geq \frac{\kappa}{\log(\frac{1}{\delta})\log(T)}$. Therefore, $T \geq \max\{2(136D^3S\sqrt{A\kappa})^8, 2(D^3S\sqrt{A})^{16}\} = 2\max\{(136D^3S\sqrt{A\kappa})^8, S^{12}A^3D^2, DSAlog^2(T)\kappa, D^2SA\kappa, D^2S^2\log(T)\kappa\}$.

### D.1    Proof of Lemma 16

In Algorithm 3, we maintain two learning process. We use $I_{x,y}(t)$ to indicate whether the $t$-th step is contained by the first process. For $t \geq T_0 + 1$, we set $I_{x,y}(t) = 0$. Let $M_1$ be the MDP with transition probability $P^{(x,y)}$ and reward $\mathbf{1}_y$, and $h^{(1)}$, $\rho^{(1)}$ denote the optimal bias function and the optimal average reward of $M_1$ respectively. In the same way we define $M_2$, $h^{(2)}$ and $\rho^{(2)}$ according to transition probability $P^{(y,x)}$ and reward $\mathbf{1}_x$.

For the first process, the regret $\mathcal{R}^{(1)} = \sum_{1\leq t\leq T_0, I_{x,y}(t)=1} \rho^{(1)} + \sum_{1\leq t\leq T_0, s_{t+1}=y, I_{x,y}(t)=1}(\rho^{(1)} - 1) = (t^{(1)} + k^{(1)})\rho^{(1)} - k^{(1)}$, where $t^{(1)} = \sum_{1\leq t\leq T_0} I_{x,y}(t)$ and $k^{(1)} = |\{t \leq T_0 | s_{t+1} = y, I_{x,y}(t) = 1\}|$. We aim to prove that with probability $1 - p$ for some $p \in (0, 1)$, it holds that

$$|\mathcal{R}_1| \leq 34DS\sqrt{AT_0\gamma}. \tag{43}$$

Because $\rho^{(1)} = \frac{1}{D_{xy}+1}$, assuming (43) holds, we have $|\frac{t^{(1)}}{k^{(1)}} - D_{xy}| \leq \frac{68D^2 S\sqrt{AT_0\gamma}}{k^{(1)}}$. On the other side, we define $t^{(2)} = \sum_{1 \leq t \leq T_0}(1 - I_{x,y}(t))$, $k^{(2)} = |\{t \leq T_0 | s_{t+1} = x, I_{x,y}(t) = 0\}|$, and thus $\mathcal{R}_2 = (t^{(2)} + k^{(2)})\rho^{(2)} - k^{(2)}$. Assuming

$$|\mathcal{R}_2| \leq 34DS\sqrt{AT_0\gamma} \qquad (44)$$

holds, it follows that $|\frac{t^{(2)}}{k^{(2)}} - D_{yx}| \leq \frac{68D^2 S\sqrt{AT_0\gamma}}{k^{(2)}}$. Noticing that $|k^{(1)} - k^{(2)}| \leq 1$ and $t^{(1)} + t^{(2)} = T_0$, we derive that $k^{(1)} \geq \frac{T_0}{2D}$ and $k^{(2)} \geq \frac{T_0}{2D}$. Therefore, we get that

$$|\frac{t^{(1)}}{k^{(1)}} - D_{xy}| \leq \frac{68D^2 S\sqrt{AT_0}}{k^{(1)}} \leq \frac{136D^3 S\sqrt{A\gamma}}{\sqrt{T_0}}$$

$$|\frac{t^{(2)}}{k^{(2)}} - D_{yx}| \leq \frac{68D^2 S\sqrt{AT_0}}{k^{(2)}} \leq \frac{136D^3 S\sqrt{A\gamma}}{\sqrt{T_0}}.$$

Because $\sqrt{T_0} \geq 136D^3 S\sqrt{A\gamma}$, we conclude that $|\frac{t^{(1)}}{k^{(1)}} - D_{xy}| \leq 1$ and $|\frac{t^{(2)}}{k^{(2)}} - D_{yx}| \leq 1$ with probability $1 - 2p$.

Theorem2 in [Jaksch et al., 2010] provides a solid foundation to prove (43) holds with high probability. Following the analysis of this theorem, we have some lemmas below.

**Lemma 17.** *Let $X_1, X_2, ...$ be i.i.d. discrete random variables with support $\mathcal{X}$. Let $I_n \in \{0, 1\}$ be random variables in $\{0, 1\}$ for $n = 1, 2, ...$. Assume that for each $n$, $X_n$ is independent of $\{I_1, ..., I_n\}$. Let $a_k = \min\{i \geq 1 | \sum_{j=1}^{i} I_j \geq k\}$. For any $k \geq 1$, if $a_k < \infty$ with probability 1, then the joint distribution of $(X_{a_1}, ..., X_{a_k})$ is the same as the joint distribution of $(X_1, ..., X_k)$, which means $X_{a_1}, ..., X_{a_k}$ are i.i.d. random variables.*

*Proof.* When $k = 1$, for each $i \geq 1$, conditioning on $a_1 = i$, the distribution of $X_{a_k}$ is the same as the distribution of $X_1$, since $X_i$ is independent of $(X_1, ..., X_{i-1}, I_1, ..., I_i)$. Because $a_k < \infty$ with probability 1, then we have $\mathbb{P}(X_{a_k} = x) = \sum_{i=1}^{\infty} \mathbb{P}(a_k = i)\mathbb{P}(X_1 = x) = \mathbb{P}(X_1 = x)$ for any $x \in \mathcal{X}$. For $n \geq 2$, we assume that this lemma holds for $k = n - 1$. In the same way we have that for any $x \in \mathcal{X}$, $\mathbb{P}(X_{a_n} = x | a_1, a_2, ..., a_n, X_1, ..., X_{a_{n-1}}) = \mathbb{P}(X_1 = x)$. It then follows that for any $(x_1, ..., x_n) \in \mathcal{X}^n$, $\mathbb{P}(X_{a_1} = x_1, ..., X_{a_n} = x_n) = \mathbb{P}(X_{a_1} = x_1, ..., X_{a_{n-1}} = x_{n-1})\mathbb{P}(X_{a_n} = x_n | X_{a_1} = x_1, ..., X_{a_{n-1}} = x_{n-1}) = \mathbb{P}(X_{a_1} = x_1, ..., X_{a_{n-1}} = x_{n-1})\mathbb{P}(X_1 = x_n) = \Pi_{i=1}^{n}\mathbb{P}(X_1 = x_i)$. Then the conclusion follows by induction. $\square$

**Lemma 18.** *With probability $1 - \frac{\delta}{60T_0^6}$, in any episode, the true transition probability $P$ is in $\mathcal{P}$.*

*Proof.* Because the rewards $\{r_{s,a}\}_{s \in \mathcal{S}, a \in \mathcal{A}}$ are assumed to be known in the beginning, it suffices to make sure $|P_{s,a} - \hat{P}_{s,a}^{(1)}|_1 \leq \sqrt{\frac{14SA\log(2AT_0/\delta_0)}{\max\{N_{s,a}^{(1)}(t), 1\}}}$.

To apply Lemma 17, we have to make sure $a_k \leq \infty$ with probability 1 for $\forall k \leq T_0$. But it's easy to see that, if we let $I_n = I_{x,y}(t(n, s, a))$ for $n \leq T_0$ where $t(n, s, a)$ is the first time $(s, a)$ is visited for $n$ times (if the visit number of $(s, a)$ is less than $n$, we set $t(n, s, a) = T_0 + 1$ and $I_n = I_{x,y}(T_0 + 1) = 0$). For $T_0 + 1 \leq n \leq 2T_0$, we set $I_n = 1$, then it follows $a_k \leq 2T_0$ for $\forall k \leq T_0$. Note that $I_{x,y}(t)$ is a function of the random events before the $t$-th round, and thus $I_{x,y}(t)$ is obviously independent of subsequent states $(s_{t+1}, s_{t+2}, ...)$. When $n \geq T_0 + 1$, $I_n$ is independent of all other random variables. As a result, for any $k \leq T_0$, the conclusion of Lemma 17 holds for $\hat{P}_{s,a,1}, \hat{P}_{s,a,2}, ...$ and $I_1, I_2, ...$, where $\hat{P}_{s,a,i} \in \mathbb{R}^S$ is the result of the $i$-th try of executing $a$ in $s$.

Because $N_{s,a}^{(1)}(t) \leq T_0$, according to Lemma 17, the distribution of $\hat{P}_{s,a}^{(1)}(t)$ is the same as the distribution of $\frac{1}{N_{s,a}^{(1)}(t)}\sum_{i=1}^{N_{s,a}^{(1)}(t)} P_{s,a,i}$, where $P_{s,a,1}, P_{s,a,2}, ...$ are i.i.d. distributed obeying multinomial distribution with parameter $P_{s,\cdot}$. Based on the analysis in Lemma 17 [Jaksch et al., 2010], we conclude that with probability $1 - \frac{\delta}{60T_0^6}$, , for any $t \leq T_0$ and any $(s, a)$, it holds that

$$|P_{s,a} - \hat{P}_{s,a}^{(1)}(t)| \leq \sqrt{\frac{14SA\log(2AT_0/\delta_0)}{\max\{N_{s,a}^{(1)}(t), 1\}}}$$

$\square$

**Lemma 19.** *Let $P'_k$ denote the transition model of the optimal extended MDP in the $k$-th episode, and $u_k$ denote the optimal bias function of $mdp(P'_k, \mathbf{1}_y)$. Then we have $sp(u_k) \leq D_y := sup_{z \neq y} D_{zy}$.*

*Proof.* Firstly, it's easy to see that $u_{k,y} \geq u_{k,z}$ for any $z \in \mathcal{S}$. Assume that there exists $z$ such that $u_{k,y} - u_{k,z} > D_y \geq D_{zy}$. We can design a nonstationary policy to achieve better value for $u_{k,z}$: in the first, we start from $z$ following some policy to reach $y$ as quickly as possible. Because the true transition model $P \in \mathcal{P}$ in each episode, we can reach $y$ within $D_{zy}$ steps in expectation. After reaching $y$, we follow the original optimal policy. Let $R_t(s)$ be the optimal $t$-step accumulative reward starting from $s$ and $\rho$ be the corresponding optimal average reward. According to the definition of optimal bias function, we have $lim_{t \to \infty} R_t(z) - \rho t = u_{k,z} \geq lim_{t \to \infty} R_{t-D_{zy}}(y) - \rho t \geq u_{k,y} - D_{zy}$. Therefore, $sp(u_k) \leq \max_z \{u_{k,y} - u_{k,z}\} \leq D_{zy}$. $\qquad\square$

According to the derivation in Section 4 [Jaksch et al., 2010], we have that

$$\mathcal{R}(mdp(P^{(x,y)}, \mathbf{1}_y), T_0) \leq |\sum_k v_k^T (P'_k - I)^T u_k| \leq |\sum_k v_k^T (P_k - I)^T u_k| + |\sum_k v_k^T (P'_k - P_k) u_k|$$

$$\leq D\sqrt{\frac{5}{2} T \log(\frac{8T_0}{\delta_0})} + DSA \log_2(\frac{8T}{SA}) + (2D\sqrt{14S \log(\frac{2AT_0}{\delta_0})} + 2)(\sqrt{2} + 1)\sqrt{T} \tag{45}$$

holds with probability $1 - 2T_0 \frac{\delta}{12T_0^{5/4}} - \frac{\delta}{60T_0^6}$.

**Remark:** We can prove (45) holds with high probability for all $t \leq T_0$ in the same way. As a result, we conclude that, with probability $1 - 3SAT_0^2 \delta$, for any $t \leq T_0$, it holds that $\mathcal{R}(mdp(P^{(x,y)}, \mathbf{1}_y), t) \leq 34DS\sqrt{AT_0\gamma}$.

With a slight abuse of notations, we use $reg_{s,a}$ to denote the single step regret for $mdp(P^{(x,y)}, \mathbf{1}_y)$. Noting that $sp(h^{(1)}) = \frac{D_y}{1 + D_{xy}} \leq D$, according to (19) in Lemma 12, for any $t \leq T_0$ it holds that

$$\mathcal{R}(mdp(P^{(x,y)}, \mathbf{1}_y), t) - \sum_{i=1}^t reg_{s_i, a_i} \geq -2\sqrt{T_0\gamma}D - D \geq -34DS\sqrt{AT_0\gamma}$$

with probability $1 - \delta$. Therefore, we conclude that with probability $1 - 4SAT_0^2 \delta$, it holds that $|\mathcal{R}(mdp(P^{(x,y)}, \mathbf{1}_y), t)| \leq 34DS\sqrt{AT_0\gamma}$ for any $t \leq T_0$.

**Algorithm 3** LD: Learn the Diameter

**Input:** $T_0, \delta_0, x \neq y \in \mathcal{S}$

$t \leftarrow 1, I_{x,y}(t) \leftarrow 0, t_{lu}^{(1)} \leftarrow 1, t_{lu}^{(2)} \leftarrow 1, \pi^{(1)}(s), \pi^{(2)}(s) \leftarrow$ arbitrary policy, $\forall s$;

$N_{s,a}^{(1)}(t) \leftarrow 0, N_{s,a}^{(2)}(t) \leftarrow 0, N_{s,a,s'}^{(1)}(t) \leftarrow 0, N_{s,a,s'}^{(2)}(t) \leftarrow 0 \ \hat{P}_{s,a,s'}^{(1)}(t) \leftarrow 0, \hat{P}_{s,a,s'}^{(2)}(t) \leftarrow 0,$
$\forall s, a, s'$;

**if** current state is not $x$ **then**
$\quad r^{(t)} \leftarrow \mathbf{1}_x$;
**else**
$\quad r^{(t)} \leftarrow \mathbf{1}_y$;
**end if**
**for** $t = 1, 2, ... T_0$ **do**
$\quad$ **if** $r^{(t)} = \mathbf{1}_x$ **then**
$\quad\quad I_{x,y}(t) \leftarrow 0$;
$\quad\quad$ **if** $\exists (s, a)$, s.t. $N_{s,a}^{(1)}(t) \geq 2N_{s,a}^{(1)}(t_{lu}^{(1)})$ or $t = 1$ **then**
$\quad\quad\quad t_{lu}^{(1)} \leftarrow t$;
$\quad\quad\quad$ update $\mathcal{P}$ as: $\mathcal{P} = \{P' | \forall (s,a), |P'_{s,a} - \hat{P}_{s,a}^{(1)}(t)|_1 \leq \sqrt{\frac{14SA\log(2AT_0/\delta_0)}{\max\{N_{s,a}^{(1)}(t),1\}}}$
$\quad\quad\quad P_1 \leftarrow \arg\max_{Q \in \mathcal{P}} \rho(mdp(Q^{(x,y)}, \mathbf{1}_x))$;
$\quad\quad\quad \pi^{(1)} \leftarrow$ optimal policy for $mdp(P_1^{(x,y)}, \mathbf{1}_x)$;
$\quad\quad$ **end if**
$\quad\quad$ Execute $\pi^{(1)}(s_t)$, get $r_t = r^{(t)}(s_t, a_t)$ and transits to $s_{t+1}$;
$\quad\quad$ **if** $s_{t+1} = x$ **then**
$\quad\quad\quad r^{(t+1)} = \mathbf{1}_y$
$\quad\quad$ **end if**
$\quad$ **else**
$\quad\quad I_{x,y}(t) \leftarrow 1$;
$\quad\quad$ **if** $\exists (s, a)$, s.t. $N_{s,a}^{(2)}(t) \geq 2N_{s,a}^{(2)}(t_{lu}^{(2)})$ or $t = 0$ **then**
$\quad\quad\quad t_{lu}^{(2)} \leftarrow t$;
$\quad\quad\quad$ update $\mathcal{P}$ as: $\mathcal{P} = \{P' | \forall (s,a), |P'_{s,a} - \hat{P}_{s,a}^{(2)}(t)|_1 \leq \sqrt{\frac{14SA\log(2AT_0/\delta_0)}{\max\{N_{s,a}^{(2)}(t),1\}}}$
$\quad\quad\quad P_2 \leftarrow \arg\max_{Q \in \mathcal{P}} \rho(mdp(Q^{(y,x)}, \mathbf{1}_y))$;
$\quad\quad\quad \pi^{(2)} \leftarrow$ optimal policy for $M_2'$;
$\quad\quad$ **end if**
$\quad\quad$ Execute $\pi^{(2)}(s_t)$, get $r_t = r^{(t)}(s_t, a_t)$ and transits to $s_{t+1}$;
$\quad\quad$ **if** $s_{t+1} = y$ **then**
$\quad\quad\quad r^{(t+1)} = \mathbf{1}_x$
$\quad\quad$ **end if**
$\quad$ **end if**
$\quad$ Update:
$\quad N_{s,a}^{(1)}(t+1) = \sum_{i=1}^t I[s_t = s, a_t = a, r^{(t)} = \mathbf{1}_x]; N_{s,a}^{(2)}(t) = \sum_{i=1}^t I[s_t = s, a_t = a, r^{(t)} = \mathbf{1}_y]$
$\quad N_{s,a,s'}^{(1)}(t+1) = \sum_{i=1}^t I[s_t = s, a_t = a, s_{t+1} = s', r^{(t)} = \mathbf{1}_x]; N_{s,a,s'}^{(2)}(t+1) = \sum_{i=1}^t I[s_t = s, a_t = a, s_{t+1} = s', r^{(t)} = \mathbf{1}_y]$;
$\quad \hat{P}_{s,a,s'}^{(1)}(t+1) = \frac{N_{s,a,s'}^{(1)}(t+1)}{\max\{N_{s,a}^{(1)}(t+1),1\}}; \hat{P}_{s,a,s'}^{(2)}(t+1) = \frac{N_{s,a,s'}^{(2)}(t+1)}{\max\{N_{s,a}^{(2)}(t+1),1\}}$.
**end for**
**Return:** $\left(\frac{|\{t|r_t=\mathbf{1}_y\}|}{|\{t|s_t=y,r^{(t-1)}=\mathbf{1}_y\}|}, \frac{|\{t|r_t=\mathbf{1}_x\}|}{|\{t|s_t=x,r^{(t-1)}=\mathbf{1}_x\}|}\right)$.

## Footnotes

[6]We use $\mathbf{1}_y$ to denote the vector $v$ satisfying $v_s = I[s = y], \forall s$.