[Reviews · NeurIPS 2019]

Reviewer 1



The paper focuses on the important problem of designing optimal algorithms for exploration-exploitation (whose upper-bound matches the lower bound). The paper is not well organized and written. It is difficult to abstract from the mathematical formulation and grasps the key ideas behind the improvement of the regret bound. As far as I understood, the first important component in improving the bound is to use variance dependent confidence intervals (ie Bernstein). Together with the knowledge of H, this allows designing a tighter optimism (Eq. 6 in Alg 2) resulting in a \sqrt{HS} gain when bounding the term P'-P ie 3k. This idea is not novel and has been used several times in the literature for regret minimization. The oldest reference I can remember is (Lattimore and Hutter, 2012). As mentioned also in their paper, the problem is that there is no algorithmic solution known when this additional constraint is considered. This is one of the limitations of the proposed algorithm (as also mentioned by you). Moreover, I'm not sure that the optimization problem you would like to solve is well posed. Fruit et al. showed that the REGAL.D problem is not well posed in general (and it seems to me you are considering the same problem). The fact you mentioned you try to consider the relaxation proposed by Fruit et al. shows that you are aware of this issue but there is no explicit mention in the paper. Is the problem well defined? In line 4 you consider deterministic policies. In light of what shown by Fruit et al., are you sure that it is enough to consider deterministic policies? The second important part for proving the improved regret bound is to carefully bound the term (\hat{P} -P)(h_k-h^*). It is not clear how this result is obtained. The key component for this is Lemma 1 but I failed in understanding how this result for flat MDPs is used in the proof of Thm 1. In general, the proofs are difficult to follow since are not supported by informal arguments and high-level intuitions are missing. For example, Ortner (2008) provided an example showing that h_k may not converge to h^*. Here is not clear why this can be obtained in general. The only reason I can think of is when a minimum number of samples (that is polynomial in the MDP parameters) is available in each state-action pair. As far as I understood this is somehow the idea behind estimating the diameter. However, there is no discussion or explanation of this in the paper. To conclude, I cannot guarantee that Thm 1 is correct since I do not understand the important steps of the proof. Note also that the best known bound for UCRL2 with Bernstein is \sqrt{D \Gamma S A T} and has been proved by Fruit et al. 2019 as part of the tutorial at ALT 2019. You approach is thus improving by a factor \sqrt{\Gamma}. Moreover, you should mention the work of (Talebi and Maillard, 2018) that provides a problem dependent upper-bound using the environmental norm. I'm familiar with this literature (I've been working on similar approaches) and the paper is very cryptical to me. The authors mainly focus on the final results rather than providing intuitions of the techniques (eg mathematical tools) necessary to achieve such results. Finally, even in the case, the contribution is technically correct (the problem is well-posed and the proof is correct), this is not closing the question of designing optimal algorithms since EBF is not implementable. Line2 89-91: There is a reference to Eq 2 that should be 1 Definition 4 is not intuitive. You should provide a high-level idea (something more than just call it to count of arrivals) Line 137: You introduce the single step regret reg_{s,a}. this term is not common in the normal analysis but it can be expressed as negative advantage function (reg_{s,a} = -A(s,a)), right? This will provide a direct interpretation to the reader. Lines 150-153: you should explain where the modified version of (2) is coming from. In general, it is not easy to understand the utility of the flat MDP for the analysis of EBF. I didn't fully understand it even after reading the appendix. Lattimore and Hutter, PAC Bounds for Discounted MDPs, 2012 Ortner, Online Regret Bounds for Markov Decision Processes with Deterministic Transitions, ALT 2008 Fruit, Pirotta and Lazaric, Improved Analysis of UCRL2B, Technical Report, Tutorial at ALT 2019 [https://rlgammazero.github.io/] Talebi and Maillard, Variance-aware regret bounds for undiscounted reinforcement learning in mdps, ALT 2018 After feedback: thanks for the valuable feedback. It is now more clear what is the term you aim to bound (that is not |h^* - h_k|) but it is not clear the intuition/tools that enables you to save the \sqrt{\Gamma}. While I think that closing the gap between upper and lower bound is an important problem, it is also important to understand the tools/ideas enabling this improvement. What you call reg_sa is called optimality gap in RL (ie advantage function) (see Burnetas and Katehakis. Optimal adaptive policies for markov decision processes. 1997). If I understood correctly the idea is to use the optimality gap to make the MDP "flat". Can you explain better how this is used to design the algorithm? Can you provide a formal statement about the fact that the optimization is well-posed? Notes about the mistake in optimistic PSRL can be also found here: Qian et al. Concentration Inequalities for Multinoulli Random Variables. Technical Report, Tutorial at ALT 2019 [https://rlgammazero.github.io/docs/JFPL2018notesPSRL.pdf]

Reviewer 2



Thanks the authors for the feedback. After a long discussions with the other reviewers, I maintain the positive review. But I agree with reviewer 1 in the sense that the authors should put significant amount of efforts in improving the presentation of the main results. Specifically, a crystal clear high level roadmap for the results should be added. -------------------------------------------------Original Comments----------------------------------- It appears to me that the contribution of the work is pretty solid. One (possibly minor) technical question, is the algorithm EBF computationally efficient? Is it possible to leverage EVI alike technique to efficiently execute this step? The quality of writing needs to be improved. For example, 1. Definition 4 is not sound, it is required that all te_k and ts_k to be defined w.r.t. L. 2. in line 142, why would the sum of reg be O(\sqrt{N})? Many more can be found throughout the paper. There is also some misleading parts in the related works section, for example, it has been pointed out in many places that [Bartlett and Tewari 09] contains some mistakes.

Reviewer 3



I do understand that it is important to demonstrate the existence of algorithms achieving the regret lower bound while they may require some non-trivial cost or assumption. However, my major concern is the intractable complexity of the proposed algorithms, and thus the absence of the evaluation in canonical cases. As the lower bound is defined in a mini-max sense, achieving the mini-max lower bound does not guarantee efficiency in canonical cases with moderate sub-optimality gap. Hence, it would be helpful if a performance guarantee in terms of the sub-optimality gap is provided. Minor comments/questions: - In my understanding, Jaksch et al.'s lower bound is built upon no prior knowledge on the bias function. Is it possible to obtain similar lower bound under the assumption that the upper bound analysis requires? - According to [Agarwal and Jia 17], Lemma C.1 is an extension of results in [Osband and Van Roy 16]. Does it mean the result of [Osband and Van Roy 16] is untrue? - Is it possible to translate the upper bounds for the finite-horizon MDPs? - line 38: impossible - line 89: wrong reference? - Have you contacted the authors of [Agarwal and Jia 17] about the errors in their lemmas? I'm not very sure if someone else's mistake should be pointed out in this way.

[Author Response · NeurIPS 2019]

We thank the reviewers for constructive comments and questions.

One common major concern of the reviewers is whether EBF can be efficiently implemented. We have been thinking
about this problem since the submission deadline and have made substantial progress in designing a practical version
of EBF algorithm. To solve the optimization problem in line5 Algorithm1 efficiently, we relax some constraints (e.g.
$\pi_k$ is deterministic), and apply EVI alike techniques to compute the constrained optimal policy in the extended MDP.
However, we have not finished the details completely before the rebuttal deadline. As a result, we have to give up
adding the implementation of EBF algorithm to this submission.

Other concerns are answered as below. The typos, wrong references, missing references and unclear expressions like
Def. 4 would be fixed carefully in the final version (if possible):

**To reviewer1**:
1. About intuition: we can catch more information about $h^*$ from the history trajectory (See line1 Algorithm2). One
important difference to previous methods is that, the order of samples in the trajectory matters in EBF algorithm, while
previous methods only use $N_{s,a}$ and $N_{s,a,s'}$ to build confidence set. As a result, $\mathcal{H}$ is a tighter confidence set for $h^*$,
which enables us to prove ④ and part of ③ are lower order terms. (See Lemma5 and Appendix.C.5)

2. We do not bound $|h_k - h^*|_\infty$. Indeed, it suffices to bound $N_{s,a,s'}^{(t_k)}|\delta^*_{s,s'} - \delta_{k,s,s'}|$ up to $\tilde{O}(\sqrt{T})$ and this is exactly
what we do.

3. We will mention the literature which first uses Bernstein's inequality to bound the uncertainty.

4. An MDP is flat iff $r_{s,a} + P^T_{s,a}h^* = h^*_s + \rho^*$ for any $s,a$. We use the notation $reg_{s,a}$ because we regard it as a
generalization of $\Delta_a = \mu^* - \mu_a$ in multi-armed bandit (MAB) problem. We will explain these expressions more
clearly.

5. As for the usage of Lemma1 in the proof of Theorem1, we apply Lemma1 to the virtual MDP with increased reward
function. Although the original MDP $M$ might not be flat, the new MDP is flat after increasing $r_{s,a}$ by $reg_{s,a}$. (See
Appendix.C.5)

6. You mean REGAL.C rather than REGAL.D? We can search among the MDPs with constant gains so that the problem
is well-posed, although it is intractable in practice.

7. The modified version of (2) refers to line1 in Algorithm2.

8. We are sorry that we are unaware of the state of the art. Consequently, we only improve an $\sqrt{S}$ (or $\sqrt{\Gamma}$) factor
compared to the work you mentioned. Nevertheless, to out best of knowledge, this is the first upper bound which matches
the lower bound with logarithm factors ignored. We will also mention [Talebi and Maillard, 2018]. In our analysis,

the dominant term in regret is $\sum_{s,a}\sqrt{N_{s,a}^{(T)}V(P_{s,a},h^*)} \leq \sqrt{\sum_{s,a}N_{s,a}^{(T)}\sum_{s,a}V(P_{s,a},h^*)} = \sqrt{T\sum_{s,a}V(P_{s,a},h^*)}$,

which outperforms the result in [Talebi and Maillard, 2018] by at least an $\sqrt{S}$ factor.

**To reviewer3**:
1. We will add a reference of upper bound of $\tilde{O}(\sqrt{N})$ in line 142.
2. We will check the related works section carefully. We will mention the works in the comments of reviewer1. The
analysis about REGAL.C [Bartlett and Tewari 09] is correct, although that paper contains some other mistakes.

**To reviewer4**:
1. You mean a problem-dependent regret bound of $O(poly(S,A,H)\sum_{s,a}\log(T)/reg_{s,a})$ like the regret bound of
$O(\sum_a \log(T)/\Delta_a)$ in the MAB problem? We can prove this regret bound is unreachable in the worst case where some
state $s$ has $o(T)$ visit count. Under the assumption the visit count (in expectation) of each state $s$ is at least $CT$ for
some conditional number $C$, our method for estimating the bias function works, and thus it is hopeful get a regret bound
of $O(poly(S,A,H,\frac{1}{C})\sum_{s,a}\log(T)/reg_{s,a})$. However, this assumption seems too strong for undiscountted RL.

2. In the case $H$ is known, the example for lower bound of $\Omega(\sqrt{SATH})$ was proposed in [Bartlett and Tewari, 2009],
although the authors claimed a wrong lower bound.

3. Yes. There is a similar mistake in the proof of Lemma3 [Osband and Van Roy 16].

4. Yes. But regret bound of $\tilde{O}(\sqrt{SATH})$ has been proved for finite horizon MDP with efficient algorithms (e.g. [Azar
et al.2017].

5. We have sent an email to the authors, but did not get replies before the rebuttal deadline. We ensure that this issue
will be carefully dealt with according to their suggestion.

[Meta-Review · NeurIPS 2019]

This paper has lead to a long and thoughtful discussion between the reviewers. The main points that were raised are the following: + The results are novel and close a long-standing gap between upper and lower bounds in a very important problem. - The presentation is subpar in that even a very well-versed expert of the topic had trouble verifying not only the proof details, but also the high-level intuition of the analysis. While the reviewers have agreed that the results are significant and they definitely bring the field forward, an expert reviewer argued that the step forward is perhaps not significantly big enough to warrant publication in the present form. However, after much discussion, the other reviewers made a strong case for acceptance and all reviewers agreed that the community would clearly benefit from this paper being published. That said, I strongly encourage the authors to work hard on improving the presentation for the final version.